# Inner-layer Token Self-modulation as Another Scaling Axis for LLMs

Yebin Yang [1][2]  Huaijin Wu [1]  Jingtao Han [2]  Yu Wang [2]  Xiaohan Qin [1]  Jingzhi Wang [1]
Debing Zhang [2]  Junchi Yan [1]

## Abstract

LLMs have traditionally scaled along dense dimensions, where performance is coupled with near-linear increases in computational cost. While MoE decouples capacity from compute, it introduces large memory overhead and hardware efficiency challenges. To overcome these, we propose token-indexed parameters as a novel, orthogonal scaling axis that decouple model capacity from FLOPs. Specifically, we introduce ReToken and MoRT, which augment Transformer layers with modulation vectors retrieved from auxiliary embedding tables. These vectors modulate the backbone via lightweight, element-wise operations, incurring negligible FLOPs overhead. Extensive experiments on both dense and MoE backbones, spanning from 190M to 9.8B parameters, demonstrate that our approach consistently reduces validation loss and significantly improves downstream task performance (e.g., +7.3 on ARC-C, +6.3 on GSM8K). Rigorous isoFLOPs analysis further confirms that MoRT fundamentally shifts the quality–compute Pareto frontier, achieving comparable model quality with 35% less compute relative to vanilla MoE architectures, and we validate that token-indexed parameters exhibit a predictable power-law scaling behavior. Moreover, our efficient implementation ensures that the overhead introduced by ReToken and MoRT remains marginal.

## 1. Introduction

The development of large language models (LLMs) is closely linked to the scaling laws of Transformer architectures (Vaswani et al., 2017). The conventional approach to enhancing model performance involves increasing the number of parameters and training tokens, which typically yields smooth power-law improvements (Kaplan et al., 2020; Hoffmann et al., 2022). However, both FLOPs and GPU memory requirements scale approximately linearly with model size, while the availability of high-quality text data is becoming increasingly limited. As a result, simply scaling dense models leads to diminishing marginal returns and may even cause performance degradation in data-constrained settings (Villalobos et al., 2024; Kim et al., 2025).

To address the inefficiencies of dense scaling, Mixture-of-Experts (MoE) architecture (Fedus et al., 2022; Dai et al., 2024) has emerged as a promising alternative, decoupling model capacity from computation by leveraging sparsely activated expert subnetworks while maintaining approximately constant active FLOPs. However, since the relationship between loss and sparsity follows a log-linear trend (Tian et al., 2025), the benefits of expert sparsity also saturate rapidly. Moreover, sparse models also exhibit lower sample efficiency, requiring larger datasets to reach convergence (Krajewski et al., 2024; 2025), alongside significant engineering challenges to ensure hardware efficiency and low latency (Huang et al., 2024a; Liu et al., 2024b) for routing balance (Kim et al., 2024).

Recognizing that scaling dense parameters, data, and expert sparsity all encounter fundamental bottlenecks, in this paper, we explore token-indexed parameters as an orthogonal and complementary scaling dimension. Specifically, we propose a module called ReToken, which augments each Transformer layer by applying a token-specific modulation vector—retrieved from learned embedding tables—to gate the MLP residual (He et al., 2016) via lightweight Hadamard products. Furthermore, we extend ReToken into a sparse variant, Mixture of ReToken (MoRT), which maintains a pool of modulators per token and employs a lightweight router to select context-appropriate mixtures.

This architectural design explicitly addresses the bottlenecks inherent in the aforementioned scaling paradigms. First, by relying on element-wise modulation instead of dense matrix multiplications, ReToken injects substantial capacity with minimal FLOPs overhead, thereby overcoming the efficiency limits of dense scaling. Second, unlike standard MoE models that incur high communication and HBM overhead, our retrieval-based mechanism is largely decoupled

---

[1]School of AI, Shanghai Jiao Tong University [2]Xiaohongshu Inc.. Correspondence to: Junchi Yan <yanjunchi@sjtu.edu.cn>.

*Proceedings of the 43rd International Conference on Machine Learning*, Seoul, South Korea. PMLR 306, 2026. Copyright 2026 by the author(s).

from the backbone computation. This enables asynchronous prefetching of parameters, which can be overlapped with backbone execution—effectively mitigating the latency typically associated with high sparsity.

Extensive experiments reveal that MoRT fundamentally shifts the quality-compute Pareto frontier, delivering consistent performance gains even as the backbone model scale increases. Furthermore, we explicitly validate the scalability of this dimension: the loss exhibits a log-linear trend with the number of token-indexed parameters, confirming that this axis follows a scaling law similar to dense parameters. **To summarize, our contributions are as follows:**

- We introduce token-indexed parameters as a novel scaling axis that expands model capacity without increasing FLOPs, offering a complementary dimension to traditional dense and sparse scaling.

- We propose ReToken and MoRT, practical architectures that leverage token-indexed modulation to enhance both dense and MoE models with negligible FLOPs overhead.

- We show a 35% compute saving in iso-compute scaling analysis and consistent downstream gains (e.g., ARC_C: +7.3; MMLU: +7.1; GSM8K: +6.3 on a 3.2B MoE backbone), validating the effectiveness of our approach.

- We propose efficient implementation strategies with which, training throughput loss is less than 7%; for inference, there is no additional GPU memory footprint, with a moderate latency increase of $\leq 7.3\%$.

## 2. Related Works

**Scaling Law.** The Kaplan scaling law (Kaplan et al., 2020) empirically shows that LLM performance follows power-law with parameters, data, and compute, enabling extrapolation across orders of magnitude. Chinchilla (Hoffmann et al., 2022) refined these findings into compute-optimal training prescriptions, arguing that many earlier LLMs were under-trained and that optimal scaling under fixed training compute requires increasing training tokens roughly proportionally with model parameters. Sardana et al. (2023) incorporate deployment considerations by explicitly accounting for inference cost in the scaling objective, showing that when inference demand is very high, the overall-optimal strategy may favor smaller models trained for longer to minimize end-to-end compute.

In parallel, scaling-law frameworks have been extended to Mixture-of-Experts (MoE) architectures. Krajewski et al. (2024) introduced a granularity hyperparameter for MoE models and found MoE consistently outperforms compute-matched dense models, with gains increasing at larger scales. Tian et al. (2025) proposed the efficiency leverage metric and empirically linked efficiency gains to factors such as expert-activation fraction via power laws, predicting that well-configured MoE can match dense performance with

substantially less compute. Overall, these works systematize LLM scaling laws across compute allocation, inference cost, and alternative architectures, offering practical guidance for improving efficiency.

**Vocabulary Scaling** emerges as another factor for quality and efficiency. Tao et al. (2024) provides a compute-optimal rule indicating that larger models benefit from larger vocabularies and show fixed-FLOPs gains. In continual-training scenarios, Takase et al. (2024) show that replacing the old vocabulary with a better-matched one outperforms keeping the original tokenizer. Huang et al. (2025) decouples input/output vocabularies to enlarge only the input side with no extra inference cost; SuperBPE (Liu et al., 2025) extends BPE with a simple pretokenization curriculum—learn subwords first, then allowing merges across whitespace to form multi-word tokens—improving encoding efficiency and downstream performance. BLT (Pagnoni et al., 2025) replaces fixed-vocabulary tokens with dynamically sized byte patches as the main computation units, enabling tokenizer-free scaling with improved efficiency and robustness.

*Distinction from our approach:* While existing methods primarily scale the vocabulary dimension V—often utilizing techniques like hash N-grams (Huang et al., 2025; Yu et al., 2025)—they are inherently constrained by combinatorial limits. Such expansions tend to capture fixed, local surface patterns without deepening semantic understanding. In contrast, we scale along the feature dimension $d$, providing a high-dimensional, context-interactive space in which tokens can acquire richer semantics through attention-mediated interactions during training.

**See Appendix A for more related works.**

## 3. Methodology

### 3.1. Preliminary

3.1.1. PRE-NORM TRANSFORMER BLOCK

We adopt the Pre-Norm Transformer (Wang et al., 2019) as our backbone, a prevalent architecture in current SOTA open-source models (Yang et al., 2025; Liu et al., 2024b) due to its robustness in stabilizing training dynamics and gradients (Zhu et al., 2024). In this setting, each sub-module operates on a normalized hidden state and contributes an additive residual update. Concretely, for token $x$ at layer $\ell$,

$$\Delta\mathbf{a}_x^\ell = \text{Attn}^\ell\big(\text{RMSNorm}(\mathbf{h}_x^\ell)\big), \qquad (1)$$

$$\tilde{\mathbf{h}}_x^\ell = \mathbf{h}_x^\ell + \Delta\mathbf{a}_x^\ell, \qquad (2)$$

$$\Delta\mathbf{m}_x^\ell = \text{FFN}^\ell\big(\text{RMSNorm}(\tilde{\mathbf{h}}_x^\ell)\big), \qquad (3)$$

$$\mathbf{h}_x^{\ell+1} = \tilde{\mathbf{h}}_x^\ell + \Delta\mathbf{m}_x^\ell, \qquad (4)$$

where $\text{FFN}^\ell(\cdot)$ is the MLP for dense model and becomes sparsely-activated experts for MoE.

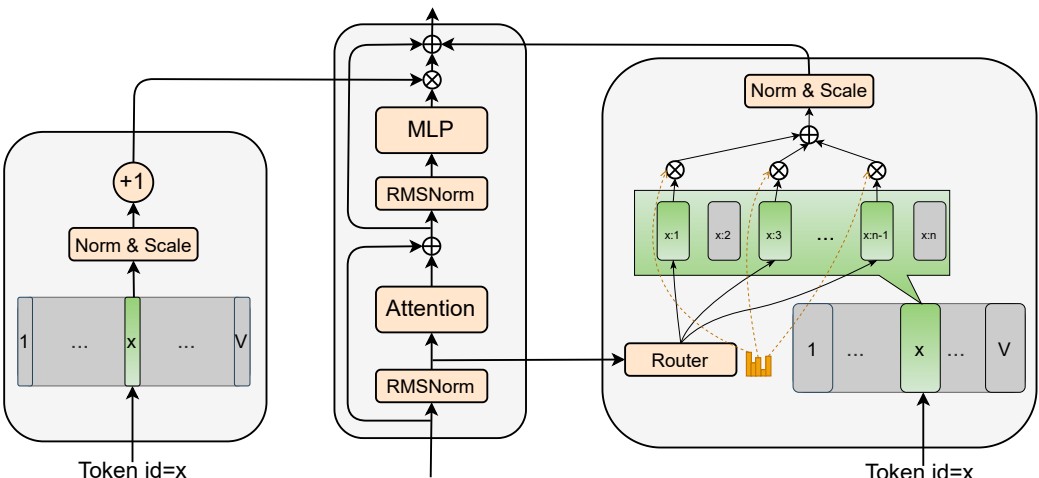

*Figure 1.* **Architecture of ReToken/MoRT.** ReToken (left) augments each Transformer layer with a token-indexed table. Each token retrieves a modulation vector, applies norm and a learnable per-dimension scaling, and forms a lightweight multiplicative gate to modulate the FFN update via element-wise products. MoRT (right) generalizes ReToken by maintaining a pool of token-indexed modulators and using a router conditioned on the hidden state to select a sparse Top-$K$ mixture per token; the mixed modulator is normalized+scaled and injected as an additional residual alongside the backbone update. Both are plug-in bypass modules implemented with table lookups and element-wise operations, allowing retrieval to be overlapped with backbone and adding negligible compute.

### 3.1.2. SCALING LAWS AND ISOFLOP PROFILES

**Scaling Law Form.** Neural scaling laws provide an empirical yet principled lens through which to reason about the quality-compute trade-off of LLM training. According to the Kaplan scaling law (Kaplan et al., 2020), the test loss can be modeled as:

$$\mathcal{L}(N_c, D) = \left[ \left( \frac{A}{N_c} \right)^{\frac{\alpha}{\beta}} + \frac{B}{D} \right]^{\beta}, \quad (5)$$

where $N_c$ denotes the compute-intensive parameters (in MoE cases, it refers to activated parameters), excluding both embedding and LM prediction head; $D$ is the number of training tokens, and $A, B, \alpha, \beta$ are empirical constants.

**IsoFLOPs Profiles.** IsoFLOPs profiles provide an empirical procedure for estimating the compute-optimal allocation between model size and data under a fixed training FLOPs budget (Hoffmann et al., 2022). Under standard Transformer FLOPs accounting, training a model with $N_c$ compute-intensive parameters[1] for $D$ tokens costs approximately $C \approx 6 N_c D$ FLOPs (Pearce & Song, 2024). The compute-optimal loss at budget $C$ is defined as

$$\mathcal{L}^*(C) = \min \mathcal{L}(N_c, D), \quad s.t.\ 6 N_c D = C \quad (6)$$

To estimate $\mathcal{L}^*(C)$ in practice, a set of target budgets $\{C_i\}$ is fixed. For each $C_i$, model size $N_c$ is swept over a grid, and the corresponding token budget is set to $D = C_i/(6 N_c)$ so that each run matches the same total FLOPs. After training, the final held-out loss is recorded for each configuration,

yielding an IsoFLOPs curve of loss versus $N_c$ at constant $C_i$, which typically exhibits a U-shaped profile with a clear minimum. A quadratic fit around the valley can be used to estimate the loss-optimal model size $N_c^*(C_i)$ and token count $D^*(C_i)$ for each budget. Collecting minima across budgets forms the empirical efficient frontier $\{(C_i, \mathcal{L}^*(C_i))\}$.

### 3.2. ReToken

ReToken scales the token embedding along hidden dimension $d$. As illustrated in Fig. 1 (left), every transformer layer $\ell$ keeps a learnable table $\mathbf{E}^\ell \in \mathbb{R}^{V \times d}$. Each token retrieves a vector $\mathbf{E}^\ell[x]$ with its ID $x \in [V]$ and gates the backbone module's residual updates via Hadamard products.

**Mechanism.** Let $x$ be the token id, $\Delta \mathbf{m}_x^\ell$ be the MLP module increments. ReToken forms multiplicative gate $\mathbf{p}_x^\ell \in \mathbb{R}^d$:

$$\mathbf{p}_x^\ell = \mathbf{1} + \mathbf{s}^\ell \odot \mathrm{Norm}_\varepsilon(E^\ell[x]), \quad (7)$$

where $\mathbf{s}^\ell \in \mathbb{R}^d$ is a learnable per-dimension scaler and $\mathrm{Norm}_\varepsilon(\mathbf{u}) = \frac{\mathbf{u}}{\|\mathbf{u}\|_2 + \varepsilon}$ ($\varepsilon$ is a small constant to avoid division by zero). We empirically validate the effectiveness of this normalization term in Appendix E.2. The gated MLP increments is

$$\Delta \hat{\mathbf{m}}_x^\ell = \Delta \mathbf{m}_x^\ell \odot \mathbf{p}_x^\ell, \quad (8)$$

which then adds to the backbone residual:

$$\mathbf{h}_x^{\ell+1} = \tilde{\mathbf{h}}_x^\ell + \Delta \hat{\mathbf{m}}_x^\ell. \quad (9)$$

### 3.3. Mixture of ReToken (MoRT)

To further unlock the potential of token-indexed parameters, we introduce MoRT. MoRT generalizes from a static mapping to a dynamic, context-aware routing framework. By

---

[1]consist of attention qkvo matrices and mlp parameters (or activated experts for MoE)

maintaining a larger pool of modulators and leveraging the hidden state as a contextual signal to guide the selection, MoRT adaptively selects a sparse mixture of parameters for each token instance. This design enables the model to capture context-dependent semantics and significantly expands the parameter space while preserving the efficiency of sparse retrieval. As shown in Fig. 1 (right).

**Mechanism.** Each token is equipped with a pool of $n_e$ modulators per layer and uses a router to pick top-$K$ of them given the hidden state.

Formally, for token $x$, let $\mathrm{RMSNorm}(\mathbf{h}_x^\ell)$ be the input of attention in layer $\ell$. Each layer maintains a pool $\{\mathbf{E}_i^\ell \in \mathbb{R}^{V \times d}\}_{i=1}^{n_e}$. A linear router computes logits

$$\mathbf{g}_x^\ell = (\mathrm{RMSNorm}(\mathbf{h}_x^\ell))^\top \mathbf{R}^\ell \in \mathbb{R}^{n_e}, \ \mathbf{R}^\ell \in \mathbb{R}^{d \times n_e}, \tag{10}$$

selects $G_x^\ell = \mathrm{TopK}(\mathbf{g}_x^\ell, K)$, and forms normalized weights $w_i^\ell = \frac{\sigma(g_i^\ell)}{\sum_{j \in G_x^\ell} \sigma(g_j^\ell)}$ for $i \in G_x^\ell$ with a Sigmoid $\sigma$ (Nguyen et al., 2024). The mixed token-indexed vector is:

$$\mathbf{e}_x^\ell = \sum_{i \in G_x^\ell} w_i^\ell \, \mathbf{E}_i^\ell[x] \in \mathbb{R}^d. \tag{11}$$

We normalize and apply a learnable element-wise scaler $\mathbf{s}_{\mathrm{MoRT}}^\ell \in \mathbb{R}^d$, producing the additive residual injection. To ensure the variance of the hidden states controllable (Radford et al., 2019), we scale with an extra factor $\frac{1}{\sqrt{2N_l}}$, where $N_l$ is the number of layers of the backbone. The ablation of this scaling factor is provided in Appendix E.1.

$$\Delta \mathbf{r}_x^\ell = \frac{1}{\sqrt{2N_l}} \cdot \mathbf{s}_{\mathrm{MoRT}}^\ell \odot \mathrm{Norm}_\varepsilon(\mathbf{e}_x^\ell), \tag{12}$$

Finally, $\Delta \mathbf{r}_x^\ell$ is fused into the layer write-back together with MLP output $\Delta \mathbf{m}_x^\ell$:

$$\mathbf{h}_x^{\ell+1} = \tilde{\mathbf{h}}_x^\ell + \Delta \mathbf{m}_x^\ell + \Delta \mathbf{r}_x^\ell \tag{13}$$

To encourage all the embedding experts to be adequately utilized and trained, we incorporate an auxiliary load-balancing loss during training, similar to standard practice in MoE models. Implementation details are provided in Appendix C.

### 3.4. System Efficiency

Token-indexed parameters shift the main system concern from GEMMs to parameter access and memory footprint. While the Transformer backbone is typically compute-bound, token-indexed modules have negligible FLOPs but are dominated by memory access. Accordingly, our system design targets:

1. keeping token-indexed access off the GEMM critical path by overlapping retrieval with backbone compute via asynchronous prefetch/overlap (Sec. 3.4.3), and minimizing memory traffic via token deduplication (Sec. 3.4.2);

2. reducing the HBM footprint of token-indexed tables via embedding model parallelism (Sec. 3.4.4) and CPU offloading (Sec. 3.4.5).

These strategies allow ReToken/MoRT to scale to massive memory capacities with minimal impact on efficiency.

### 3.4.1. COMPUTATIONAL COMPLEXITY ANALYSIS
Per token and per layer, ReToken adds a normalization, a Hadamard product for gating, and a residual write-back, all of which are $\mathcal{O}(d)$.[2]

MoRT additionally computes router logits and mixes top-$K$ modulators. Routing is a single matrix-multiply with cost $\mathcal{O}(dn_e)$, and mixture application is $\mathcal{O}(Kd)$. In practice $n_e$ and $K$ are small constants, so the overall overhead remains linear in $d$ and is negligible compared to the backbone's $\Theta(d^2)$ attention/FFN computes.

### 3.4.2. MEMORY ACCESS AND ZIPFIAN PATTERNS
The dominant overhead of ReToken/MoRT is memory traffic from table lookups. For each layer, ReToken reads $d$ elements per token, while MoRT reads $Kd$. However, token frequencies follow a Zipfian distribution (Zipf, 2016), so high-frequency tokens are repeatedly accessed. This enables *token deduplication*: unique token ids in a micro-batch are gathered once and then scattered back to all occurrences, avoiding redundant reads.

### 3.4.3. DECOUPLING AND ASYNCHRONOUS OVERLAP

ReToken/MoRT are bypass modules, decoupled from backbone, so their execution can be scheduled independently of attention/FFN computation, enabling overlapping memory access with compute. Concretely, embedding gathers for some layer can be issued asynchronously while the backbone executes GEMMs for the current layer, and the retrieved vectors are fused only at the layer write-back (Eqs. (9) and (13)). As a result, a substantial portion of the retrieval latency can be hidden under compute, making the end-to-end throughput impact small in practice.

### 3.4.4. EMBED-PARALLEL IN TRAINING
Token-indexed tables are large but sparsely accessed, making them well-suited for *embedding model parallelism*. Tables can be sharded across GPUs, reducing per-device memory footprint, allowing larger micro-batches and increasing training throughput. For ReToken, the cross-device communication per token is $d$ elements. For MoRT, a naive implementation would communicate $Kd$ elements, but this can be avoided by *owner-rank premixing*: the device that owns the selected modulators performs the weighted sum locally, and only the mixed vector ($d$-sized) is communicated.

---

[2]From the roofline (Williams et al., 2009) perspective, these element-wise Ops could be limited by memory traffic and kernel-launch. Kernel fusion can amortize launches and avoid extra memory round-trips.

### 3.4.5. CPU OFFLOADING IN INFERENCE

Since token-indexed parameters are accessed sparsely, the host-to-device transfer volume depends on the number of requested vectors, not on the full table size. Per layer, transferring the retrieved values scales as $\mathcal{O}(d)$ for ReToken and $\mathcal{O}(Kd)$ for MoRT, independent of $V$ and of the overall parameter capacity stored on the host. This property makes CPU offloading a practical option during inference to save HBM usage. Specifically, the tables can reside in CPU memory, while only the small set of vectors needed by the current batch are streamed to GPUs and overlapped with backbone execution.

### 3.5. Scaling Hypothesis of Token-Indexed Parameters

Building on the Kaplan scaling law (Eq. 5) and the FLOPs constraint $C \approx 6N_cD$ introduced in the Preliminary, we introduce one core assumption: effective parameter count $N_{\text{eff}}$. This assumption integrates token-indexed parameters into the scaling law and yields a *scale-invariant* isoperformance compute saving in the compute-optimal regime.

**Core assumption: effective parameters $N_{\text{eff}}$.** We denote the backbone compute-intensive parameters as $N_c$ and the token-indexed parameters as $N_n$. Let $\eta \triangleq N_n/N_c$ be the parameter expansion ratio. MoRT retrieves and injects token-indexed vectors per token and per layer; its usable capacity depends on routing sparsity. We capture this with a MoRT architecture-dependent discount function $\gamma(\rho)$, where $\rho$ denotes the MoRT activation sparsity ($\rho = K/n_e$). We define

$$N_{\text{eff}} \triangleq N_c + \gamma(\rho)N_n = N_c\big(1 + \eta\gamma(\rho)\big). \quad (14)$$

Importantly, $N_{\text{eff}}$ characterizes *effective capacity*, whereas the dominant training FLOPs are still governed by $N_c, D$.

**Incorporating $N_{\text{eff}}$ into the Kaplan scaling law.** Starting from Eq. 5, we replace the $N_c$ with $N_{\text{eff}}$:

$$\mathcal{L}_{\text{MoRT}}(N_c, D; \eta, \rho) = \left[\left(\frac{A_{\text{MoRT}}}{N_c}\right)^{\frac{\alpha}{\beta}} + \frac{B}{D}\right]^{\beta}, \quad (15)$$

where $A_{\text{MoRT}} \triangleq A/(1+\eta\gamma(\rho))$. This highlights that MoRT is equivalent to rescaling the constant in the model-size term, while leaving the data term $B/D$ and the dominant FLOPs form unchanged.

**Isoperformance compute saving under compute-optimal training.** Let $\mathcal{L}^*(C)$ denote the compute-optimal efficient frontier of backbone model defined in Eq. 6. Similarly, the MoRT compute-optimal frontier becomes a *multiplicative downward shift*:

$$\mathcal{L}^*_{\text{MoRT}}(C; \eta, \rho) = \big(1 + \eta\gamma(\rho)\big)^{-\frac{\alpha\beta}{\alpha+\beta}} \cdot \mathcal{L}^*(C). \quad (16)$$

Together, for any target test loss $\mathcal{L}^\star$, the minimal compute required satisfies

$$C^\star_{\text{MoRT}}(\mathcal{L}^\star) = \frac{1}{1 + \eta\gamma(\rho)} C^\star_{\text{base}}(\mathcal{L}^\star). \quad (17)$$

The key insight of Eq. 17 is that the **compute saving ratio is independent of the backbone scale** and depends only on MoRT architectural hyperparameters (e.g., $\eta$, $\rho$, and $\gamma(\cdot)$). The detailed derivations of Eq. 16 and Eq. 17 are provided in Appendix B.

## 4. Experiments

### 4.1. Main Quality Results

We first validate that token-indexed parameters consistently improve quality across diverse architectures and backbone scales, before examining scaling behaviors and efficiency.

**Backbone Models.** We establish backbone using both dense and MoE architectures. For dense models, we use four Qwen-style (Yang et al., 2025) models of varying sizes: 190M(S), 0.5B(M), 1B(L), and 1.5B(XL) parameters. For MoE models, we use two highly-sparse configurations: one with 250M activated parameters and 1.5B total parameters (1.5B-A250M), and another with 0.5B activated parameters and 3.2B total parameters (3.2B-A0.5B). Each MoE layer contains 145 experts, with one shared expert (Dai et al., 2024) and 144 routed experts, from which the top-8 are activated per token. For more details about the model configurations, please refer to Appendix F.

**ReToken/MoRT Configuration.** We evaluate two methods: ReToken and MoRT. Both are implemented as attached modules that augment the backbone model without modifying its architecture. ReToken is benchmarked on dense-XL and both MoE models. MoRT is evaluated on the two MoE backbones. For MoRT, we set the number of modulator experts to $n_e = 5$ and activate the top $K = 2$.

**Datasets and Training.** All models are pretrained with Megatron-LM framework (Shoeybi et al., 2019). The S, M, and L dense models are pretrained for 100B tokens on the Fineweb-edu (Lozhkov et al., 2024), an open-source collection of text dataset. The XL dense model and all MoE models are trained on another high-quality dataset curated from online corpora, which includes general text, code, math, and multilingual content after rigorous filtering. Both XL dense and 1.5B-A250M MoE are trained for 300B tokens, and the 3.2B-A0.5B MoE for 500B tokens. All training configurations for the ReToken and MoRT models are kept identical to those of their corresponding backbones. For detailed training hyperparameters, see Appendix F.

**Evaluation.** We evaluate downstream benchmark performances of dense-XL, 1.5B-A250M and 3.2B-A0.5B MoEs with OpenCompass (Contributors, 2023) framework in 4 domains: *Knowledge* (MMLU (Hendrycks et al., 2020), TriviaQA (Joshi et al., 2017), ARC (Clark et al., 2018), GPQA (Rein et al., 2024)), *Reasoning* (Hellaswag (Zellers et al., 2019), C3 (Sun et al., 2020), BBH (Suzgun et al., 2022), SocialIQA (Sap et al., 2019)),

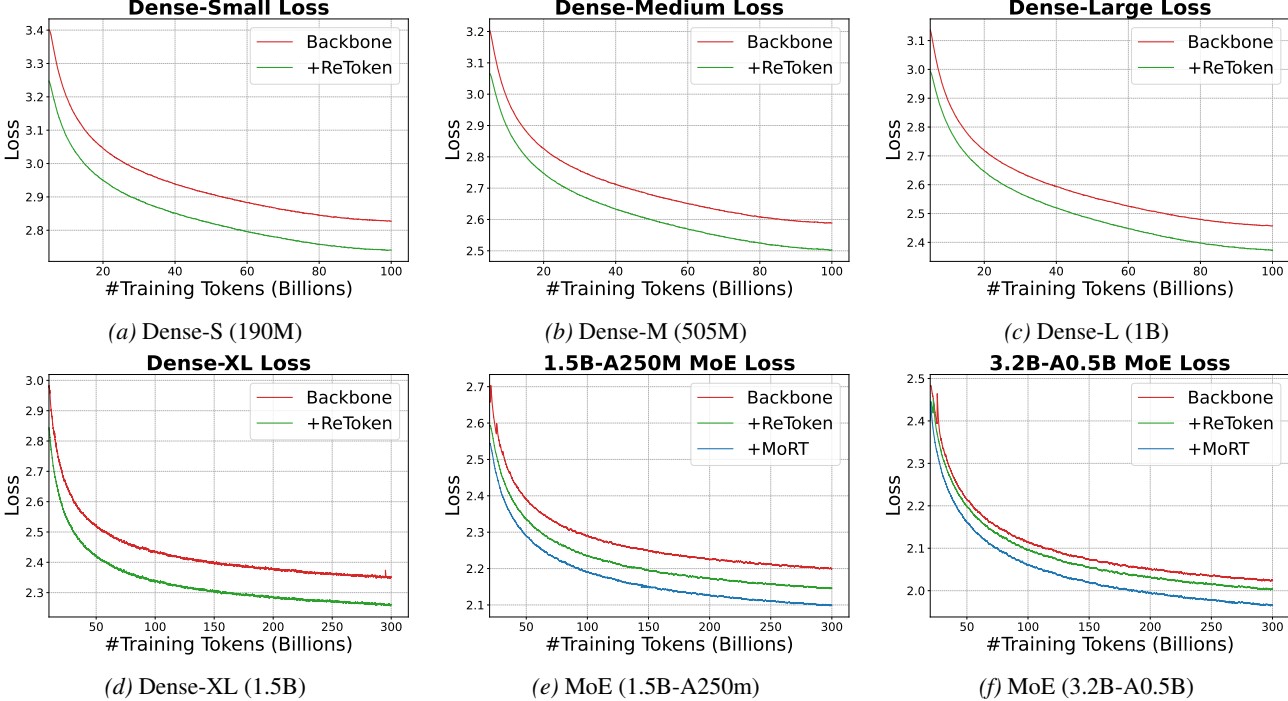

*Figure 2.* Training loss for dense and MoE backbones and corresponding ReToken and MoRT variants. The top row shows ReToken's performance on Dense-S(190M), M(505M), and L(1B) backbones. The bottom row shows results on Dense-XL (1.5B), 1.5B-A250M MoE, and 3.2B-A0.5B MoE backbones. In all settings, ReToken (and MoRT for MoE backbone) achieves a consistently and significantly lower training loss.

*Code* (MBPP (Austin et al., 2021), HumanEval (Chen et al., 2021), LiveCodeBench (Jain et al., 2024)) and *Math* (MATH (Hendrycks et al., 2021), GSM8K (Cobbe et al., 2021), DROP (Dua et al., 2019)).

### 4.1.1. RESULTS AND ANALYSIS

**Training Loss.** As shown in Fig. 2, our proposed methods achieve a consistent and significant reduction in training loss across all model scales and architectures. For the dense models, ReToken consistently maintains a lower loss trajectory. Similarly, for the MoE models, both ReToken and MoRT show a clear advantage over the vanilla backbones, reducing the training loss for both the 1.5B and 3.2B MoE configurations. This shows that ReToken and MoRT are robust methods for improving optimization and data compression.

**Downstream Performance.** The benefits observed during training translate to significant and consistent improvements in downstream tasks. As shown in Table 1, ReToken boosts the Dense-XL average accuracy from 22.22 to 26.54 (+4.32), with sizable gains on key benchmarks such as MMLU (+4.55) and TriviaQA (+9.50).

For MoE, gains are further amplified by MoRT(Table 1). On 1.5B-A250M, the average accuracy improves from 18.87 to 22.78 (+3.91); on 3.2B-A0.5B, it rises from 26.75 to 32.34 (+5.59), including strong improvements on ARC_C (+7.25) and GSM8K (+6.31).

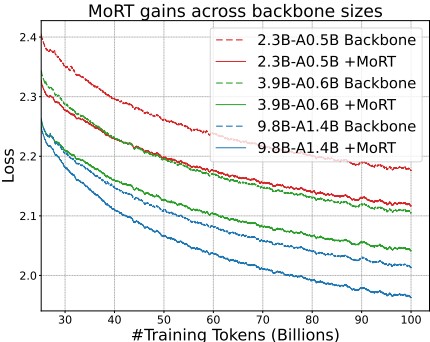

*Figure 3.* MoRT (fixed $(\eta, \rho)$) improvements remain stable and consistent across different backbone model sizes.

### 4.2. Scaling Laws

This subsection studies MoRT scalability along two axes: (i) whether MoRT's gains remain stable as the backbone scales, which is crucial for applying this feature in real-world LLMs; and (ii) what scaling behavior does it exhibit if scaling MoRT itself. Together, these results characterize token-indexed parameters as a predictable and complementary scaling dimension.

### 4.2.1. MORT IS SCALABLE TO LARGE BACKBONES

**Large-scale Pretrain** We train both MoE and +MoRT variant with three backbone sizes (2.3B-A0.5B, 3.9B-A0.6B, and A9.8B-A1.4B) under a fixed MoRT configuration $(\eta, \rho) = (50, 0.25)$ for 100B tokens.

Fig. 3 shows that MoRT's benefits are remarkably consistent

*Table 1.* Performance evaluation on comprehensive downstream tasks (Knowledge, Code, Reasoning, and Math). Backbone-only denotes the backbone without any plugin; +ReToken and +MoRT denote the same backbone augmented with the corresponding plugin. Overall Avg is the mean over all 14 tasks. The best result of each task (within the same backbone block) is highlighted in bold.

| Backbone | Configuration | Knowledge | | | | Code | | | Reasoning | | | | Math | | | Overall Avg. |
|---|---|---|---|---|---|---|---|---|---|---|---|---|---|---|---|---|
| | | MMLU | TriviaQA | ARC_C | GPQA | MBPP | HumanEval | LCB | Hellaswag | C3 | BBH | SocialIQA | MATH | GSM8K | DROP | |
| **XL** (Dense) | Backbone-only | 32.74 | 25.72 | 35.74 | 25.60 | 9.53 | 9.15 | 0.87 | 51.19 | 39.34 | 21.04 | 41.64 | 2.17 | 5.76 | 10.55 | 22.22 |
| | + ReToken | **37.29** | **35.22** | **41.58** | **29.80** | **12.20** | **9.76** | **2.61** | **55.07** | **47.83** | **24.12** | **45.97** | **3.36** | **9.07** | **17.71** | **26.54** |
| **1.5B-A250M** (MoE) | Backbone-only | 27.91 | 22.81 | 29.21 | 25.59 | 7.00 | **10.57** | 1.57 | 45.70 | 27.95 | 18.96 | 34.44 | 1.96 | 3.87 | 6.60 | 18.87 |
| | + ReToken | 30.98 | 27.31 | 30.24 | 28.28 | 10.60 | 9.96 | 1.45 | 48.01 | 29.86 | 18.82 | 36.39 | 3.04 | **4.17** | 7.99 | 20.51 |
| | + MoRT | **34.07** | **36.53** | **34.72** | **30.03** | **12.50** | 10.55 | **1.64** | **52.06** | **30.22** | **20.66** | **38.51** | **4.37** | 4.15 | **8.90** | **22.78** |
| **3.2B-A0.5B** (MoE) | Backbone-only | 36.39 | 39.57 | 39.76 | 29.47 | 19.20 | 16.46 | 2.52 | 57.05 | 36.84 | 21.87 | 42.63 | 7.16 | 13.65 | 11.86 | 26.75 |
| | + ReToken | 40.05 | 41.13 | 43.87 | 28.79 | 21.80 | 18.90 | 4.70 | 58.52 | 44.05 | 24.74 | 44.83 | 8.30 | 16.38 | 13.27 | 29.24 |
| | + MoRT | **43.47** | **46.29** | **47.01** | **31.70** | **24.51** | **19.89** | **5.96** | **60.37** | **49.01** | **28.58** | **47.15** | **10.42** | **19.96** | **18.44** | **32.34** |

across model scales. MoRT reduces the final loss by 0.059 on the 2.3B-A0.5B backbone, by 0.064 on the 3.9B-A0.6B backbone, and by 0.051 on the A9.8B-A1.4B backbone, corresponding to relative improvements of 2.7%, 3.0%, and 2.55%, respectively.

**Rigorous IsoFLOPs Validation** In Sec. 3.5, we predict that token-indexed parameters yield a *scale-invariant* improvement under compute-optimal training: across compute budgets, the efficient frontier is shifted downward by a constant multiplicative factor (Eq. 16). Here we demonstrate the validity of this hypothesis via isoFLOPs experiments.

Following the isoFLOPs protocol described in Sec. 3.1.2, we establish the efficient frontier for MoE (top-8 routing, one shared expert, and 145 total experts). We choose five FLOPs budgets that are log-uniformly spaced, $C \in \{3e18, 1e19, 3e19, 1e20, 3e20\}$, and then fit a quadratic around the minimum of the resulting U-shaped curve to obtain the compute-optimal configuration $(N_c^*(C), D^*(C))$ and its best held-out loss $\mathcal{L}^*(C)$, as shown in Fig. 4.

For each compute-optimal backbone configuration, we attach the MoRT module while keeping the backbone architecture and training setup unchanged. MoRT adds token-indexed parameters but does not change the dominant FLOPs. We use a fixed MoRT configuration with parameter expansion ratio $\eta = N_n/N_c = 50$ and activation sparsity $\rho = 0.25$. This yields another set of efficient-frontier points $\{(C, \mathcal{L}_{\text{MoRT}}^*(C))\}$ corresponding to MoRT architecture.

Across all budgets, MoRT improves the compute-optimal loss, and the improvement is well-approximated by a constant multiplicative factor, consistent with Eq. 16. To make the comparison explicit, we analyze the frontiers in log-log space. Taking logarithms of Eq. 16 gives

$$\log \mathcal{L}_{\text{MoRT}}^*(C; \eta, \rho) = \log \mathcal{L}^*(C) - g(\eta, \rho), \quad (18)$$

which predicts that the vertical gap between the two frontiers in log space is $C$-invariant and depends solely on the MoRT hyperparameters through $(\eta, \rho)$. Empirically, linear fits of $\log \mathcal{L}^*(C)$ and $\log \mathcal{L}_{\text{MoRT}}^*(C)$ versus $\log C$ yield nearly identical slopes, while MoRT exhibits a clear downward intercept shift, as shown in Fig. 5. This verifies that MoRT improves the quality-compute Pareto frontier in a

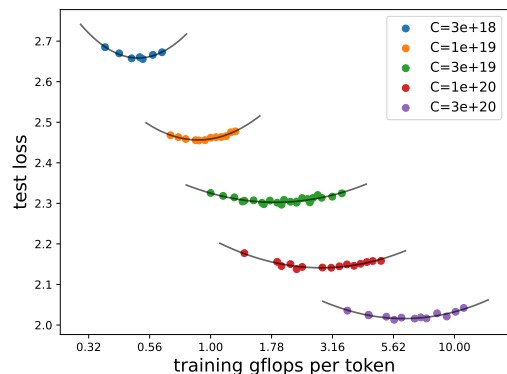

*Figure 4.* Isocompute for vanilla MoE models with compute budgets. Each curve shows test loss vs. per-token FLOPs at fixed FLOPs, with optimal points defining the efficient frontier.

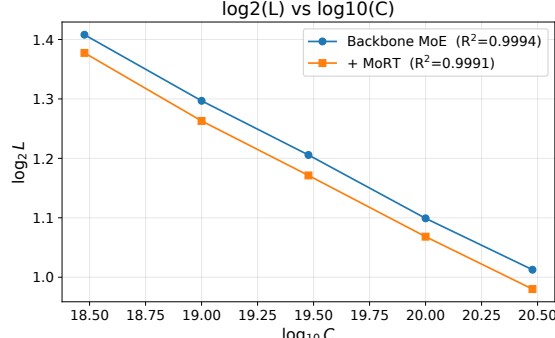

*Figure 5.* Efficient frontiers of MoE and +MoRT variant. MoRT achieves 2.27% lower loss at each compute budget, equivalent to saving 35% compute to reach each test loss.

manner that is stable across scale. Quantitatively, under matched compute budgets, MoRT achieves a consistent **2.27% loss reduction**; equivalently, to reach any target test loss, MoRT **saves 35% compute** relative to MoE. Detailed regression statistics and the mathematical derivation of the compute saving ratio are provided in Appendix D.

#### 4.2.2. SCALING MoRT ITSELF

In subsection 4.2.1, we validate that MoRT can be reliably applied to larger backbone as its relative gains remain stable under large-scale pretraining, and the rigorous IsoFLOPs study verifies a scale-invariant downward shift of the compute-optimal frontier. We now examine the complementary question: *can MoRT itself be scaled as a*

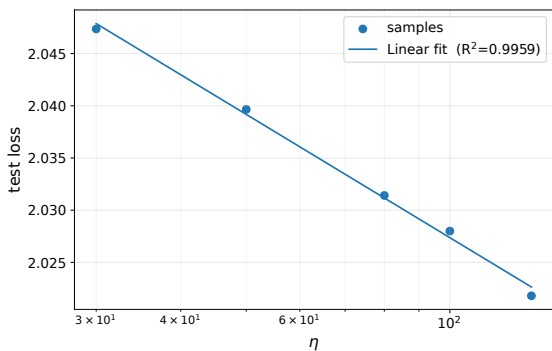

*Figure 6.* Scaling behavior of MoRT. The validation loss follows an approximately log-linear trend with respect to the MoRT parameter expansion ratio $\eta$. In particular, doubling the number of MoRT parameters reduces the loss by about **0.0118**.

*standalone capacity axis, and does it exhibit a predictable scaling behavior?*

**Experimental Setup.** We fix the backbone to a single MoE model (3.9B-A0.6B) and pretrain for 100B tokens. On top of this frozen backbone specification, we attach MoRT modules of varying capacity. Concretely, we parameterize the MoRT size using the parameter expansion ratio $\eta$; We sweep $\eta$ from 30 to 130 while fixing the sparsity ratio $\rho = 0.25$. All other MoRT hyperparameters and all training hyperparameters are kept identical across runs, so that the observed differences are attributable solely to scaling $N_n$.

**Results** Fig. 6 shows that increasing MoRT capacity leads to a clear and consistent reduction in held-out loss. Within the explored range, the improvement is monotonic with no sign of degradation, indicating that MoRT provides an effective capacity knob beyond backbone scaling. Moreover, the trend is well-approximated by a smooth power law (linear in log-space): each doubling of MoRT parameters reduces the loss by approximately **0.0118**. This provides evidence that token-indexed parameters form an orthogonal scaling axis that introduces negligible backbone compute.

### 4.3. System Efficiency

We evaluate the end-to-end training and inference efficiency of ReToken/MoRT. With sufficient system optimizations, the throughput drop of ReToken/MoRT can be bounded within 6.78% for training and 7.3% for inference, demonstrating that ReToken/MoRT is practical for both large-scale training and deployment.

**Experimental Setup.** (1) Training. We measure pretraining throughput (tokens/s) on 128×H800, with batch size 1024 and sequence length 8192, on the 3.2B-A0.5B MoE backbone. For MoRT, we set $\eta = 50$ and $\rho = 0.25$. We compare (i) backbone-only, (ii) +MoRT without optimization, (iii) +MoRT with embedding parallelism, and (iv) +MoRT with embedding parallelism plus token deduplication. (2) Inference. We benchmark inference on 8×H800 with

*Table 2.* Training throughput (tokens/s) on the 3.2B-A0.5B MoE backbone with MoRT ($\eta = 50$). EmbP denotes embedding parallel, and Dedup denotes token deduplication.

| Model | Throughput (tok/s) |
|---|---|
| Baseline | 4,838K |
| MoRT (naive) | 2,749K |
| MoRT (EmbP) | 4,024K |
| MoRT (EmbP+Dedup) | 4,510K (-6.78%) |

*Table 3.* Inference throughput on 8×H800 using SGLang (batch size=16, sequence length=4K) on the 3.2B-A0.5B MoE backbone. ReToken/MoRT use CPU offloading for token-indexed tables.

| Model | Prefill (tok/s) | Decode (tok/s) |
|---|---|---|
| Baseline | 363.7K | 4494 |
| ReToken (CPU-offload) | 360.8K (-0.8%) | 4290 (-4.5%) |
| MoRT (CPU-offload) | 355.2K (-2.3%) | 4166 (-7.3%) |

batch size 16 and sequence length 4K using SGLang (Zheng et al., 2024). We implement CPU-offloaded ReToken/MoRT and report prefill and decode throughput on the 3.2B-A0.5B MoE backbone.

**Results.** Table 2 shows that a naive MoRT implementation reduces training throughput mainly for two reasons: (i) the additional memory lookup and (ii) the token-indexed parameters increasing HBM usage, forcing a smaller microbatch and lower MFU. Embedding parallelism alleviates the latter by sharding tables across GPUs to reduce per-device memory footprint, while token deduplication addresses the former by removing repeated gathers for frequent token access. With joint optimizations, the training throughput reduction is controlled within 6.78%.

Table 3 shows that CPU-offloaded ReToken/MoRT incurs only small inference overhead. Prefill throughput drops by 0.8% (ReToken) and 2.3% (MoRT), while decode drops by 4.5% and 7.3%, respectively. This is expected as decode is more compute-light than prefill, leaving less backbone computation to overlap and hide the host-to-device transfer latency. Overall, the throughput reduction remains low, supporting practical deployment.

## 5. Conclusion and Outlook

We have established token-indexed parameters as a novel and complementary scaling axis for large language models. Our proposed architectures, ReToken and MoRT, consistently enhance both dense and MoE models with negligible system overhead. Extensive evaluations show that our methods significantly boost performance across a wide range of downstream tasks.

Rigorous isoFLOPs analysis confirms that MoRT fundamentally shifts the quality-compute Pareto frontier, achieving baseline-matching performance with 35% compute savings. Furthermore, we empirically show that token-indexed parameter itself exhibits a predictable power-law scaling be-

havior, establishing this dimension as a robust and scalable trajectory for future LLM advancement.

**Limitations.** The backbone scale in our experiments is still relatively small, and frontier-scale validation is left as future work. Beyond scale, our evaluation focuses on pretraining alone, so the interaction of token-indexed parameters with post-training has not yet been explored. We have also not studied how these parameters interact with inference-time compression techniques such as quantization and distillation.

## Acknowledgements

This work was in part supported by Scientific Research Innovation Capability Support Project for Young Faculty (U40) of the Ministry of Education of China (SRICSPYF-ZY2025019) and Xiaohongshu Inc.

## Impact Statement

This work aims to improve the efficiency of large language model scaling by adding token-indexed modulation parameters with minimal additional FLOPs. If effective at larger scales, the proposed method may reduce the compute, cost, and energy required to train and serve capable language models. At the same time, more efficient scaling may also lower the barrier to developing stronger models, which could amplify existing risks of LLMs, including biased or misleading generation, memorization, and misuse. These risks are not specific to our architecture, but should be considered when deploying models built with this technique.

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

## A. More Related Works

**Mixture of Experts.** MoE architectures have advanced toward sparser routing, finer-grained experts, and greater deployability. The sparsely-gated MoE of Shazeer et al. (2017) activates only a small subset of FFN experts per example via a trainable gate with an auxiliary load-balancing loss, scaling capacity to tens of billions of parameters with limited extra compute and validating conditional computation for language modeling and translation. Switch Transformer (Fedus et al., 2022) further simplifies routing by selecting a single expert per token, yielding a favorable speed-quality trade-off under fixed compute and enabling trillion-parameter scaling. Dai et al. (2024); Liu et al. (2024a) push extreme expert specialization by partitioning experts more finely and isolating shared experts for generic knowledge, improving utilization and reducing redundancy, and reporting competitive or superior performance to larger dense/MoE baselines at comparable cost. To ease deployment and accelerate inference, Jie et al. (2025) reparameterizes trained FFN experts into lookup tables for inference, retrieving precomputed expert outputs on demand to slash VRAM usage and avoid real-time expert compute, which yields dense-like latency in their setting.

**Large Memory Layer.** Large memory layers expand model capacity by decoupling parameters from compute. Product Key Memory (Lample et al., 2019) inserts a very large, sparsely accessed key-value table into the FFN, adding minimal extra FLOPs while improving language modeling at scale. PEER (He, 2024) further extends product-key style routing to sparsely select from a pool of over a million tiny experts, improving the performance-compute trade-off in a fine-grained MoE-like regime. Ultra-Sparse Memory (Huang et al., 2024b) introduces an ultra-sparse memory layer that activates only a few memory slots per token and reports reduced inference latency while maintaining performance. Memory Layers at Scale (Berges et al., 2024) shows that replacing some FFNs with trainable key-value memory layers scales to very large memory capacity and can outperform dense models using substantially more FLOPs, while also being competitive with MoE under matched budgets.

## B. Derivations for MoRT scaling laws

Here we provide detailed derivations for Eq. 16 and Eq. 17.

**Step 1: MoRT as an effective rescaling of the model-size term.** We start from the Kaplan form (Eq. 5). For MoRT, we introduce the effective parameter count

$$N_{\text{eff}} = N_c + \gamma(\rho)N_n = N_c\big(1 + \eta\gamma(\rho)\big), \qquad \eta \triangleq N_n/N_c. \tag{19}$$

Plugging $N_{\text{eff}}$ into Eq. 5 yields

$$\mathcal{L}_{\text{MoRT}}(N_c, D; \eta, \rho) = \left[\left(\frac{A}{N_{\text{eff}}}\right)^{\frac{\alpha}{\beta}} + \frac{B}{D}\right]^{\beta}$$

$$= \left[\left(\frac{A/(1 + \eta\gamma(\rho))}{N_c}\right)^{\frac{\alpha}{\beta}} + \frac{B}{D}\right]^{\beta}. \tag{20}$$

Define

$$A_{\text{MoRT}} \triangleq \frac{A}{1 + \eta\gamma(\rho)}. \tag{21}$$

Then MoRT has the same functional form as the backbone loss, except that the constant in the model-size term is rescaled from $A$ to $A_{\text{MoRT}}$.

**Step 2: Compute-optimal frontier and its dependence on $A$.** Under the standard training-FLOPs approximation $C \approx 6N_cD$ (Sec. 3.1.2), we can write $D = C/(6N_c)$ and view the loss as a function of $N_c$ at fixed $C$:

$$\mathcal{L}(N_c; C) = \left[\left(\frac{A}{N_c}\right)^{\frac{\alpha}{\beta}} + \frac{6BN_c}{C}\right]^{\beta}. \tag{22}$$

Minimizing the bracketed term over $N_c$ gives the compute-optimal frontier $\mathcal{L}^*(C)$. A standard calculation for this two-term trade-off (capacity term decays with $N_c$ while data term grows with $N_c$ under fixed $C$) yields the well-known dependence

$$\mathcal{L}^*(C) \propto A^{\frac{\alpha\beta}{\alpha+\beta}} C^{-\frac{\alpha\beta}{\alpha+\beta}}, \tag{23}$$

i.e., at compute-optimality the frontier is a power law in $C$, and its *intercept* scales as $A^{\frac{\alpha\beta}{\alpha+\beta}}$.

Since MoRT only changes $A \mapsto A_{\mathrm{MoRT}}$ while keeping the same $C$-dependence, we immediately obtain the multiplicative shift:

$$
\begin{aligned}
\mathcal{L}^*_{\mathrm{MoRT}}(C; \eta, \rho) &= \left(\frac{A_{\mathrm{MoRT}}}{A}\right)^{\frac{\alpha\beta}{\alpha+\beta}} \mathcal{L}^*(C) \\
&= \left(1 + \eta\gamma(\rho)\right)^{-\frac{\alpha\beta}{\alpha+\beta}} \cdot \mathcal{L}^*(C),
\end{aligned}
\tag{24}
$$

which is Eq. (16).

**Step 3: Isoperformance compute saving.** Eq. (16) implies that, in the compute-optimal regime, MoRT improves the frontier by a *constant* multiplicative factor independent of $C$. Let $k \triangleq \frac{\alpha\beta}{\alpha+\beta}$ and write $\mathcal{L}^*(C) = \kappa C^{-k}$ for some constant $\kappa$. Then Eq. (16) becomes

$$
\mathcal{L}^*_{\mathrm{MoRT}}(C) = (1 + \eta\gamma(\rho))^{-k} \kappa C^{-k}.
\tag{25}
$$

For any target loss $\mathcal{L}^\star$, solving $\mathcal{L}^\star = \kappa (C^*_{\mathrm{base}})^{-k}$ and $\mathcal{L}^\star = (1 + \eta\gamma(\rho))^{-k} \kappa (C^*_{\mathrm{MoRT}})^{-k}$ gives

$$
C^*_{\mathrm{MoRT}}(\mathcal{L}^\star) = \frac{1}{1 + \eta\gamma(\rho)} C^*_{\mathrm{base}}(\mathcal{L}^\star),
\tag{26}
$$

which is Eq. (17). Notably, the compute saving ratio depends only on MoRT hyperparameters through $(\eta, \rho)$ and $\gamma(\rho)$, and is independent of the backbone scale.

## C. Load-Balancing Auxiliary Loss for MoRT

Similar to MoE architectures that require balanced expert utilization for optimal throughput and learning, MoRT benefits from uniform routing across its $n_e$ embedding experts $\{E_i\}_{i=1}^{n_e}$. To encourage all embeddings are adequately trained and contribute to model performance, we incorporate an auxiliary load-balancing loss that encourages balanced routing distributions.

**Formulation.** Consider a training batch with $T$ tokens, where MoRT maintains $n_e$ embedding experts and each token routes to top-$K$ embeddings. Let $G_t \subseteq \{1, \ldots, n_e\}$ denote the top-$K$ routing set for token $t$, and $p_t^{(i)} \in [0, 1]$ represent the routing probability of token $t$ to embedding $i$.

We define the following quantities:

- **Aggregate routing probability:** $P_i = \sum_{t=1}^{T} p_t^{(i)}$ — the unnormalized sum of routing probabilities for embedding $i$
- **Actual token count:** $n_i = \sum_{t=1}^{T} \mathbf{1}\{i \in G_t\}$ — number of tokens actually routed to embedding $i$
- **Normalized routing probability:** $p_i = \frac{P_i}{T}$ — the average routing probability for embedding $i$
- **Normalized load fraction:** $f_i = \frac{n_i}{TK}$ — the fraction of total route capacity used by embedding $i$

The load-balancing auxiliary loss is formulated as:

$$
\mathcal{L}_{\mathrm{aux}} = \lambda \cdot n_e \sum_{i=1}^{n_e} p_i f_i = \lambda \cdot \frac{n_e}{T^2 K} \sum_{i=1}^{n_e} P_i n_i
\tag{27}
$$

where $\lambda$ is a hyperparameter controlling the strength of the load-balancing constraint.

**Intuition.** The term $p_i$ captures the expected routing distribution to embedding $i$, while $f_i$ measures its actual utilization. The loss in Equation 27 penalizes the co-occurrence of high routing probability and high actual load, thereby discouraging the concentration of routing on a subset of embeddings. This encourages a more uniform distribution where ideally $p_i \approx f_i \approx 1/n_e$ for all $i$, ensuring that all embeddings receive sufficient training signal and contribute effectively to model capacity.

**Implementation.** In practice, we compute this auxiliary loss per layer and average across all MoRT layers. The hyperparameter $\lambda$ is typically set to $10^{-4}$, balancing load distribution without overwhelming the primary language modeling objective. This auxiliary loss is added to the main cross-entropy loss during training and is automatically handled by the backward pass without requiring special gradient computation.

# D. Detailed Compute-Optimal Scaling Laws

In this section, we provide the detailed numerical results and linear fitting parameters for the IsoFLOPs analysis discussed in Sec. 4.2.1. We compare the compute-optimal performance of the vanilla MoE backbone against the MoRT augmented variant ($\eta = 50, \rho = 0.25$) across five orders of magnitude of compute budgets.

## D.1. Compute-Optimal Data Points

Table 4 lists the compute-optimal test losses ($\mathcal{L}^*$) for both the vanilla MoE backbone and the MoRT model at five distinct FLOPs budgets ($C$). These values correspond to the minima of the IsoFLOPs curves. The relative improvement is calculated as $1 - (\mathcal{L}^*_{\mathrm{MoRT}} / \mathcal{L}^*_{\mathrm{Base}})$.

*Table 4.* Compute-optimal test loss comparison between Vanilla MoE and MoRT across different compute budgets. The data corresponds to the extracted efficient frontiers.

| Budget ($C$) (FLOPs) | Vanilla MoE ($\mathcal{L}^*_{\mathrm{Base}}$) | MoRT ($\mathcal{L}^*_{\mathrm{MoRT}}$) | Reduction (%) |
|---|---|---|---|
| $3 \times 10^{18}$ | 2.6537 | 2.5981 | 2.10 |
| $1 \times 10^{19}$ | 2.4569 | 2.3999 | 2.32 |
| $3 \times 10^{19}$ | 2.3065 | 2.2521 | 2.36 |
| $1 \times 10^{20}$ | 2.1422 | 2.0969 | 2.11 |
| $3 \times 10^{20}$ | 2.0176 | 1.9726 | 2.23 |

## D.2. Linear Fitting of Efficient Frontiers

To characterize the scaling behavior, we perform a linear regression in the log-log space. Following the standard practice, we use $\log_{10}$ for the compute budget $C$ and $\log_2$ for the loss $\mathcal{L}$ (to align with previous work numerical scales). The relationship is modeled as:

$$\log_2(\mathcal{L}) = \alpha \cdot \log_{10}(C) + \beta \tag{28}$$

Based on the data points in Table 4, we obtain the following linear fits:

**Vanilla MoE (Backbone)**

$$\log_2(\mathcal{L}) = -0.2016 \cdot \log_{10}(C) + 5.1334, \quad R^2 = 0.9994 \tag{29}$$

**MoRT**

$$\log_2(\mathcal{L}) = -0.2009 \cdot \log_{10}(C) + 5.0954, \quad R^2 = 0.9991 \tag{30}$$

## D.3. Derivation of Compute Saving ratio

In this subsection, we provide the mathematical derivation for the 35% compute saving claim based on the fitted efficient frontiers.

**Formulation** Let the scaling laws for the baseline and MoRT models be expressed in the log-log space as:

$$\log_2(\mathcal{L}_{\mathrm{base}}) = \alpha \cdot \log_{10}(C) + \beta_{\mathrm{base}} \tag{31}$$

$$\log_2(\mathcal{L}_{\mathrm{MoRT}}) = \alpha \cdot \log_{10}(C) + \beta_{\mathrm{MoRT}} \tag{32}$$

where $\alpha$ represents the scaling slope (empirically $\approx -0.2016$) and $\beta$ represents the intercept. We assume the slopes are identical based on the empirical fits shown in Fig. 5; the $7 \times 10^{-4}$ difference between the two fitted slopes is within the regression error and is negligible for the calculation below.

**Compute Ratio for Iso-Loss** To estimate the compute saving, we determine the ratio of compute budgets ($C_{\mathrm{MoRT}}$ vs. $C_{\mathrm{base}}$) required to achieve the same target validation loss $\mathcal{L}^*$. Setting $\mathcal{L}_{\mathrm{base}}(C_{\mathrm{base}}) = \mathcal{L}_{\mathrm{MoRT}}(C_{\mathrm{MoRT}}) = \mathcal{L}^*$:

$$\alpha \cdot \log_{10}(C_{\mathrm{base}}) + \beta_{\mathrm{base}} = \alpha \cdot \log_{10}(C_{\mathrm{MoRT}}) + \beta_{\mathrm{MoRT}} \tag{33}$$

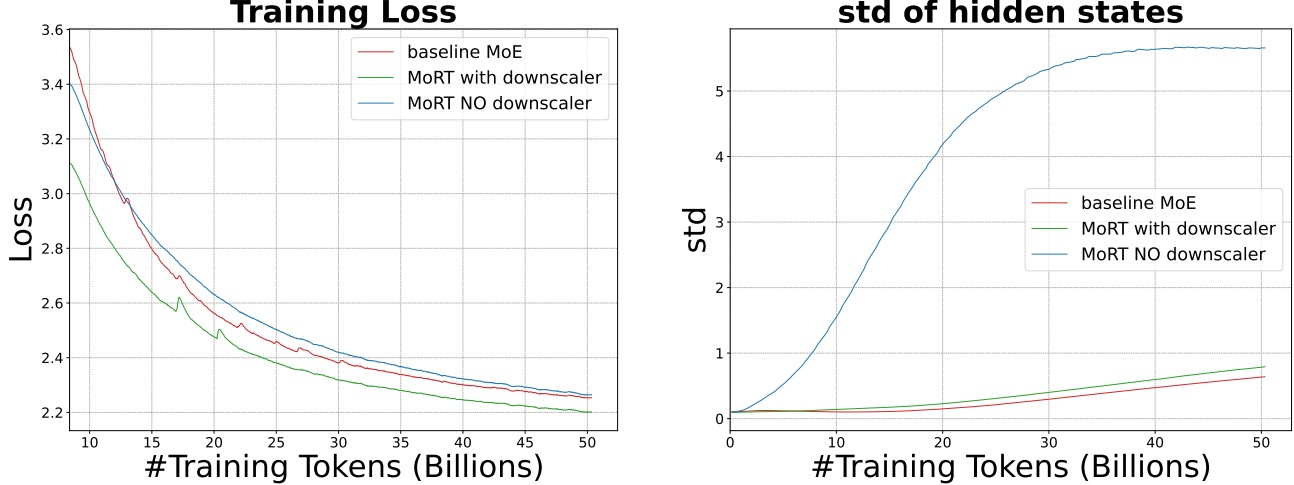

*Figure 7.* Ablation of the scaling factor $\frac{1}{\sqrt{2N_\ell}}$ in MoRT. Left: training loss. Right: averaged std of hidden states over layers and tokens.

Rearranging the terms to solve for the compute ratio:

$$\alpha(\log_{10}(C_{\mathrm{MoRT}}) - \log_{10}(C_{\mathrm{base}})) = \beta_{\mathrm{base}} - \beta_{\mathrm{MoRT}} \tag{34}$$

$$\log_{10}\left(\frac{C_{\mathrm{MoRT}}}{C_{\mathrm{base}}}\right) = \frac{\beta_{\mathrm{base}} - \beta_{\mathrm{MoRT}}}{\alpha} = \frac{\Delta\beta}{\alpha} \tag{35}$$

Thus, the compute ratio is given by:

$$\mathrm{Ratio} = \frac{C_{\mathrm{MoRT}}}{C_{\mathrm{base}}} = 10^{\frac{\Delta\beta}{\alpha}} \tag{36}$$

**Calculation**   Using the fitted parameters from our experiments:

- Slope $\alpha \approx -0.2016$

- Intercept difference $\Delta\beta = \beta_{\mathrm{base}} - \beta_{\mathrm{MoRT}} \approx 0.038$

Substituting these values:

$$\mathrm{Ratio} = 10^{\frac{0.038}{-0.2016}} \approx 10^{-0.1885} \approx 0.648 \tag{37}$$

Finally, the compute saving is:

$$\mathrm{Saving} = 1 - \mathrm{Ratio} = 1 - 0.648 \approx 35.2\% \tag{38}$$

This confirms that MoRT reduces the compute budget required to reach a target loss by approximately $35\%$.

## E. Ablation Studies

### E.1. Scaling factor $\frac{1}{\sqrt{2N_l}}$ in MoRT

To stabilize the variance of hidden states when injecting MoRT residuals, we scale the modulation update in each layer with a factor $\frac{1}{\sqrt{2N_l}}$ (Eq. 12).

We perform an ablation on the 3.2B–A0.5B MoE backbone comparing (i) the baseline(backbone MoE), (ii) MoRT *with* the scaling factor (our default), and (iii) MoRT *without* the scaling factor. All models are trained under identical data and optimization settings.

Fig. 7 (left) reports the training loss, while Fig. 7 (right) shows the standard deviation of hidden states averaged over all layers and tokens. Without the scaling factor, the activation std grows rapidly to above $5.5$ within the first 50B training

tokens and the model underperforms the baseline MoE in terms of loss. In contrast, MoRT with the proposed $\frac{1}{\sqrt{2N_\ell}}$ factor keeps variance of hidden states close to baseline in a well-behaved range and consistently achieves lower training loss than the baseline. These results confirm that the scaling factor is crucial to prevent variance explosion and to ensure that MoRT's modulation effectively improves optimization.

### E.2. Norm in ReToken/MoRT

ReToken augments each layer with a token-indexed modulation vector retrieved from an embedding table, as described in Section 3.2.

**Why normalizing the modulation vector matters.** At a high level, ReToken aims to provide *token-specific* and dimension-wise modulation. In this mechanism, the direction of $\mathbf{E}^\ell[x]$ encodes how the residual update should be modulated across hidden dimensions, whereas its magnitude primarily controls the overall strength of that modulation. Applying $\mathrm{Norm}_\varepsilon(\cdot)$ explicitly removes the magnitude degree of freedom and places the modulation vector on a hypersphere.

This has two practical benefits.

**1. Better-conditioned optimization: learning direction without chasing scale.** Without normalization, the gate depends on the raw embedding magnitude, so training needs to jointly tune both direction and norm of $\mathbf{E}^\ell[x]$ to reach an effective modulation regime. In long-horizon training with Adam-style optimizers, the effective step magnitude in parameter space is typically bounded by the scheduled learning rate (up to the factor $m/\sqrt{v}$), and in common regimes one often has $m/\sqrt{v} \approx \mathcal{O}(1)$ (Kingma, 2014; Loshchilov et al., 2024). Therefore, as $\mathbf{E}^\ell[x]$ is usually initialized at a small scale (Liu et al., 2024a), its norm tends to remain o(1) for a long time unless gradients consistently push in the radial direction. This makes it difficult for the model to quickly reach the "right" modulation strength through embedding norms alone. Normalization resolves this by decoupling direction learning from scale: the token embedding focuses on learning a direction on the hypersphere, while the learnable per-dimension scaler $\mathbf{s}^\ell$ controls the overall modulation strength. Although normalization fixes one scalar degree of freedom, it retains essentially all expressive power in high dimensions. Noting that a $d$-dimensional vector on the hypersphere still has $d - 1$ degrees of freedom.

**2. Stable and comparable modulation across tokens.** Token-indexed tables can have highly non-uniform access frequencies, which leads to heterogeneous gradient statistics across token embeddings. Without normalization, different tokens can drift to different norms, making the gate distribution highly inconsistent and potentially harming stability (e.g., some tokens barely modulate while others over-modulate). By constraining all retrieved vectors to a comparable norm, it produces a more predictable gate scale across tokens and layers, making the plugin behave like a controlled residual modulation.

**Experimental setup.** We conduct an ablation on the 3.2B-A0.5B MoE backbone, comparing three variants: (i) backbone-only, (ii) ReToken w/ norm using $\mathrm{Norm}_\varepsilon$ as above, and (iii) ReToken w.o. norm where the retrieved vector is used directly without normalization. All runs use the same architecture and training recipe; to better expose optimization differences, we intentionally *overtrain* to ∼1.3T tokens and periodically evaluate downstream accuracy throughout training. We report MMLU, ARC, CMMLU, and CEval trajectories in Fig. 8.

**Results and discussion.** Fig. 8 shows that normalization substantially improves both optimization speed and final performance. Across all four benchmarks, ReToken w/ norm rises faster in the early-to-mid training regime and consistently maintains a higher accuracy curve than ReToken w.o. norm. Moreover, the gap persists (or even widens) even under long training, indicating that normalization is not merely a transient stabilization trick but also enables ReToken to realize a higher effective capacity ceiling.

### E.3. ReToken placement: after MLP vs. after attention

ReToken is applied to the MLP residual by default. We ablate this choice by training an after-attention variant that gates the attention update instead. Both runs use a 17B-A1.5B MoE backbone trained for 87.5B tokens under identical settings.

As reported in Table 5, after-MLP outperforms after-attention by +1.08 avg, and the after-attention variant even regresses on Hellaswag. FFN outputs encode stable feature patterns that are amenable to token-type calibration, whereas attention outputs are dynamically composed contextual messages on which static rescaling may distort context integration.

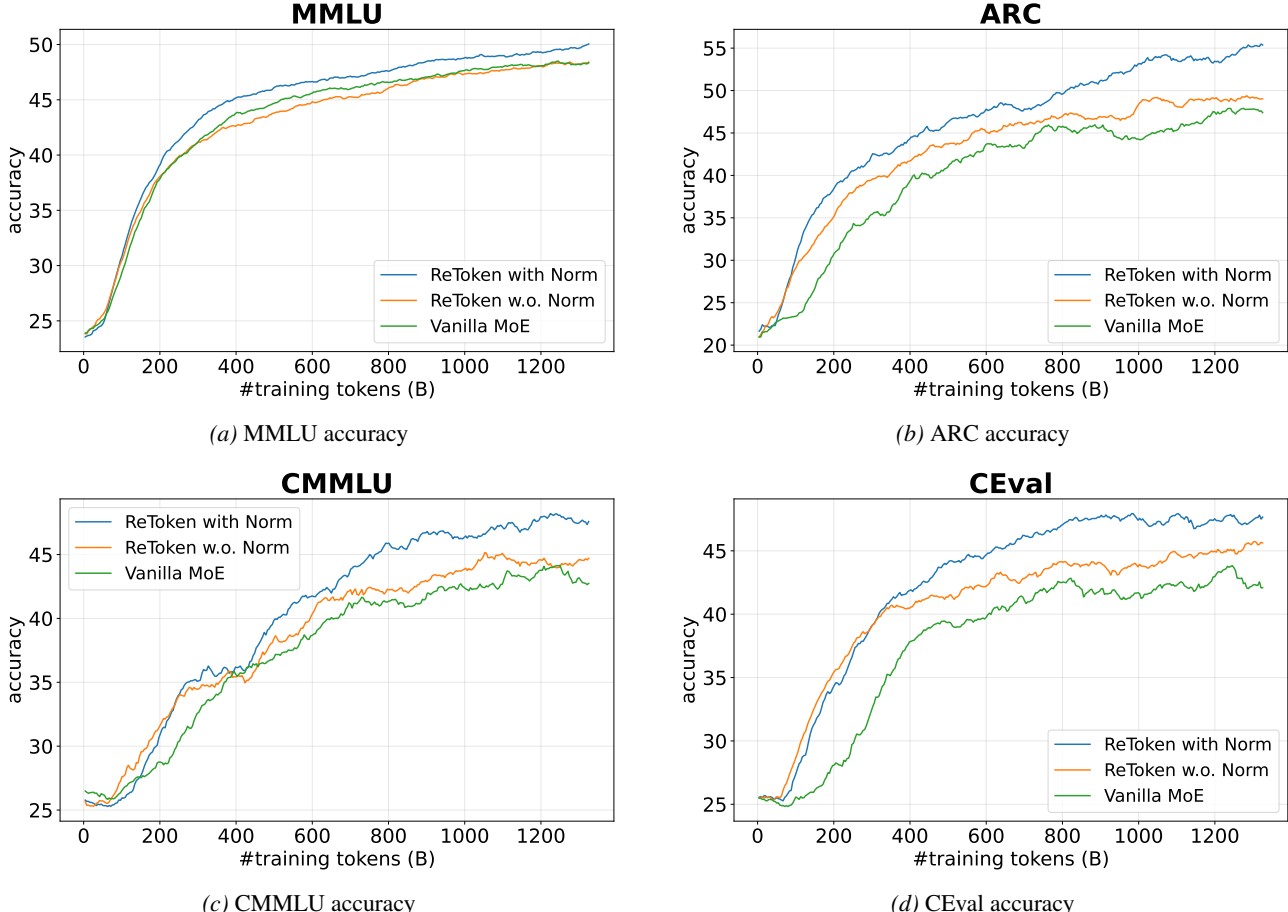

*Figure 8.* **Normalization ablation for ReToken.** Downstream accuracy trajectories on MMLU, ARC, CMMLU, and CEval during long-horizon pretraining (~1.3T tokens) on the 3.2B-A0.5B MoE backbone, comparing backbone-only, ReToken with norm, and ReToken without norm. Applying norm consistently accelerates convergence and improves the final accuracy across all benchmarks, indicating that hypersphere-normalized modulation yields more stable optimization and a higher performance ceiling.

### E.4. MoRT router input: pre-attention vs. pre-MLP

The MoRT router operates on the pre-attention hidden state by default. We ablate this against a pre-MLP routing variant that instead reads the post-attention state of the same layer. Both runs use the 17B-A1.5B MoE backbone with 36B MoRT parameters ($n_e = 4$, top-$K = 2$), trained for 87.5B tokens.

As reported in Table 6, pre-attention routing outperforms pre-MLP by $+1.17$ avg. Both inputs carry context from preceding layers; they differ only in whether the current layer's attention has been applied. Pre-attention also yields a system benefit: a larger overlap window between MoRT and the backbone computation (attention + MLP), enabling more aggressive prefetching.

### E.5. Load-balancing auxiliary loss

The auxiliary load-balancing loss (LBL, Eq. 27) prevents the MoRT router from collapsing onto a small subset of modulators. We compare otherwise-identical runs with and without this loss on the 17B-A1.5B MoE backbone, using 36B MoRT parameters ($n_e = 4$, top-$K = 2$) trained for 87.5B tokens; when present, the LBL weight is $\lambda = 10^{-4}$.

As reported in Table 7, removing LBL causes a $-3.94$ avg drop, confirming that balanced routing is critical for MoRT's effectiveness. We also monitor the load-balancing metric $n_e \sum_i p_i f_i$ (Eq. 27, without the coefficient $\lambda$) throughout training. Without LBL, the metric converges to $\sim 1.98$, indicating substantial routing concentration; with LBL, it stabilizes at $\sim 1.22$, much closer to the theoretical optimum of $1.0$ (perfect uniformity). The imbalanced routing in the no-LBL setting means a subset of embedding experts receive insufficient training signal, directly explaining the downstream degradation.

*Table 5.* ReToken placement ablation on the 17B-A1.5B MoE backbone (87.5B tokens). After-MLP is our default.

|  | Hellaswag | Xiezhi | CMMLU | MMLU | ARC-C | CEval | C3 | **Avg** |
|---|---|---|---|---|---|---|---|---|
| After attn | 34.41 | 58.98 | 49.40 | 46.95 | 48.45 | 51.20 | 64.11 | 50.50 |
| After MLP (default) | 36.71 | 59.02 | 51.88 | 48.56 | 49.14 | 50.96 | 64.77 | **51.58** |

*Table 6.* MoRT router-input ablation on the 17B-A1.5B MoE backbone (87.5B tokens, 36B MoRT, $n_e = 4$, top-$K = 2$). Pre-attention is our default.

|  | Hellaswag | Xiezhi | CMMLU | MMLU | ARC-C | CEval | C3 | **Avg** |
|---|---|---|---|---|---|---|---|---|
| Router at MLP input | 41.04 | 61.43 | 53.71 | 49.44 | 50.62 | 56.13 | 68.77 | 54.45 |
| Router at attn input (default) | 39.93 | 61.98 | 56.00 | 50.49 | 54.30 | 55.15 | 71.51 | **55.62** |

# F. Detailed Training and Model Hyperparameters

Table 8 provides the pretraining hyperparameters for dense models, covering four model sizes (small, medium, large, and XL) and their ReToken variants. Table 9 shows the pretraining hyperparameters for MoE models, specifically two configurations (1.5B-A250M, 3.2B-A0.5B) together with their ReToken and MoRT variants.

*Table 7.* Load-balancing loss ablation on the 17B-A1.5B MoE backbone (87.5B tokens, 36B MoRT, $n_e = 4$, top-$K = 2$, $\lambda = 10^{-4}$).

|          | Hellaswag | Xiezhi | CMMLU | MMLU | ARC-C | CEval | C3    | Avg   |
|----------|-----------|--------|-------|------|-------|-------|-------|-------|
| With LBL | 39.93     | 61.98  | 56.00 | 50.49 | 54.30 | 55.15 | 71.51 | **55.62** |
| No LBL   | 38.68     | 57.34  | 51.23 | 48.60 | 47.77 | 52.47 | 65.70 | 51.68 |

*Table 8.* Hyper-parameters for dense model pretraining.

|          | Parameters      | small          | medium         | large          | XL                    |
|----------|-----------------|----------------|----------------|----------------|-----------------------|
| Optimizer | lr-schedule    | cosine         | cosine         | cosine         | WSD (Wen et al., 2024) |
|          | max, min lr     | (1e-3, 1e-4)   | (8e-4, 8e-5)   | (6e-4, 6e-5)   | (6e-4, 6e-4)          |
|          | warmup-ratio    |                |                | 0.05           |                       |
|          | decay-ratio     | 0.95           | 0.95           | 0.95           | 0.00                  |
|          | optimizer       |                | AdamW (Kingma, 2014) |          |                       |
|          | weight-decay    |                |                | 0.1            |                       |
|          | grad_clip       |                |                | 1.0            |                       |
| Backbone | #params         | 190M           | 0.5B           | 1B             | 1.5B                  |
|          | hidden dim.     | 768            | 1024           | 1280           | 1536                  |
|          | #layers         | 12             | 24             | 36             | 28                    |
|          | #q heads        | 12             | 16             | 20             | 12                    |
|          | #kv heads       | 12             | 16             | 20             | 2                     |
|          | context-length  | 1024           | 1024           | 1024           | 8192                  |
|          | FFN size        | 3072           | 4096           | 5120           | 8960                  |
| Data     | vocab size      | 50304          | 50304          | 50304          | 152064                |
|          | #tokens(B)      | 100            | 100            | 100            | 300                   |
|          | batch size      | 4096           | 4096           | 4096           | 512                   |
| ReToken  | #extra params.  | 464M           | 1.2B           | 2.3B           | 6.5B                  |

*Table 9.* Hyper-parameters for MoE model pretraining.

|  | **Parameters** | **1.5B-A250M** | **3.2B-A0.5B** |
|---|---|---|---|
| Optimizer | lr-schedule | WSD | |
| | lr | 4.2e-4 | |
| | warmup-ratio | 0.05 | |
| | decay-ratio | 0.00 | |
| | optimizer | AdamW | |
| | weight-decay | 0.1 | |
| | grad_clip | 1.0 | |
| Backbone | hidden dim. | 512 | 768 |
| | #dense layers | 1 | 1 |
| | #moe layers | 11 | 17 |
| | #q heads | 8 | 16 |
| | #kv heads | 4 | 4 |
| | context-length | 8192 | 8192 |
| | #routed experts | 144 | 144 |
| | # shared experts | 1 | |
| | topK route | 8 | 8 |
| | dense FFN size | 10944 | |
| | MoE FFN size | 512 | 512 |
| Data | vocab size | 152064 | |
| | batch size | 1024 | 1024 |
| | #tokens(B) | 300 | 500 |
| ReToken | #extra params. | 934M | 2.1B |
| MoRT | $n_e$ | 5 | |
| | K | 2 | |
| | #extra params. | 4.7B | 10.5B |

