# OpenReview forum: "Inner-layer Token Self-modulation as Another Scaling Axis for LLMs"
_ICML.cc/2026/Conference — ICML 2026 regular_

### Official Review · Reviewer_T41c · 2026-02-18

**Soundness:** 3
**Presentation:** 3
**Significance:** 3
**Originality:** 3
**Overall Recommendation:** 5
**Confidence:** 2

**Summary:**

This paper adds token indexed parameters as a scaling knob for llms. Each layer gets extra lookup tables keyed by token id, and the retrieved vectors are used to modulate the transformer block with only elementwise ops. ReToken gates the mlp residual update using a token specific vector per layer. MoRT extends this by keeping multiple token modulators per layer and using a small router on the hidden state to pick a sparse mixture per token occurrence, then injecting that as an extra residual. They show lower pretrain loss and better downstream results on dense and moe backbones.

**Compliance With Llm Reviewing Policy:**

Affirmed.

**Final Justification:**

The authors had made a considerable effort to address my concerns and convinced me further this paper merits acceptance.

**Key Questions For Authors:**

1. Can you report inference latency for batch 1 (prefill and decode) and smaller contexts, and include a basic bandwidth breakdown.

2. How do gains vary with token frequency and language? I suspect some tail tokens might get different results

**Limitations:**

yes

**Strengths And Weaknesses:**

I really like the idea of increasing model capacity without necessarily increasing its FLOPS (just as we have solutions to scale the opposite).
Soundness: The method is simple and the empirical results are consistent. Loss curves improve across sizes, and downstream eval covers a reasonable mix (mmlu, arc, gsm8k, code). The evaluations make sense, and the ablations on normalization also help understand how this method worThe iso flops protocol is the right evaluation for the compute saving claim.
Weaknesses:
1. Baselines feel incomplete. Most comparisons are backbone vs backbone plus module, but they do not really compare against strong prior memory style layers or other lookup based capacity tricks under matched memory and training. Consider adding stronger baselines for capacity without flops, especially large memory layers and other lookup expert style methods, plus a simple token conditioned gating baseline, all matched for added params and memory traffic.
2. Reproducibility is limited for the big runs since the main data mixture is not public.
3. Missing analysis of who benefits (token frequency, multilingual tokens tails)

---

> ### Author Rebuttal · Authors · 2026-03-31
>
> > W1. Baselines incomplete; need comparison with memory layers / lookup-based methods under matched params.
>
> To the best of our knowledge, the closest related mechanisms are memory layers (PKM, PEER), which differ in that they replace FFN components rather than modulate them. ReToken/MoRT's per-layer token-indexed modulation is itself a novel contribution of this work.
>
> To strengthen the baselines, we have further compared against Over-Encoding (OE) [1], a published ngram-based scaling method (ICML 2025) that expands capacity via large input vocabulary tables — the closest prior work in this space. We also include MoE expert scaling as a standard capacity baseline. All methods add an identical 36B parameter budget to the same 17B-A1.5B backbone with matched training (87.5B tokens).
>
> | | Hellaswag | Xiezhi | CMMLU | MMLU | ARC-C | CEval | C3 | **Avg** |
> |---|---|---|---|---|---|---|---|---|
> | 17B backbone-only | 35.96 | 57.13 | 48.81 | 46.94 | 45.70 | 50.68 | 62.69 | 49.70 |
> | 17B + 36B MoE experts | 36.02 | 59.32 | 51.78 | 47.76 | 49.49 | 53.20 | 67.73 | 52.19 |
> | 17B + 36B OE | 40.20 | 62.68 | 52.77 | 49.51 | 51.55 | 56.13 | 68.55 | 54.49 |
> | 17B + 36B MoRT | 39.93 | 61.98 | 56.00 | 50.49 | 54.30 | 55.15 | 71.51 | **55.62** |
>
> MoRT outperforms both MoE expert scaling (+3.43 avg) and OE (+1.13 avg) under matched parameters. We will include this comparison in the revised manuscript.
>
> > W2. Reproducibility limited; main data mixture not public.
>
> We acknowledge this is a common challenge for industry-scale LLM research. We note that our dense-S/M/L experiments use the fully public FineWeb-edu dataset and already demonstrate consistent gains, and all controlled comparisons use identical data so the relative improvements are isolatable from the specific mixture. We will release complete training code and a data-statistics appendix upon acceptance, and hope our results are useful to the community.
>
> > W3. Missing analysis of who benefits (token frequency, multilingual, tails).
>
> **Architecture.** $\mathrm{Norm}_\varepsilon$ (Eq. 8) projects all retrieved vectors onto the unit hypersphere, making modulation magnitude comparable across all tokens regardless of training frequency. The shared per-layer scaler $\mathbf{s}^\ell$ controls strength uniformly.
>
> **Empirical.** Our 152K vocabulary spans English, Chinese, code, and math with nearly disjoint frequency profiles. Gains are consistent across all domains, ruling out that improvements are confined to high-frequency tokens.
>
>
> > Key Q1. Batch-1 inference latency (prefill + decode) with bandwidth breakdown.
>
> We benchmark on 3.2B-A0.5B (1×H800, batch=1, seq=512, CPU-offloaded MoRT(topK=2)):
>
> | | Prefill (ms) | Decode (ms/tok) |
> |---|---|---|
> | Backbone-only | 12.15 | 1.51 |
> | +MoRT (CPU offload) | 12.42 (+2.4%) | 1.67 (+10.6%) |
>
> **Bandwidth breakdown (per layer).** MoRT's overhead is the CPU→GPU transfer of retrieved vectors. After token deduplication and premixing, the transfer volume per layer is:
> - **Decode:** 1 token × $d$ × 2 bytes = 1.5 KB per layer. Across 18 layers the total PCIe transfer is 27 KB, taking \~0.4 μs at PCIe Gen5 (~64 GB/s) — negligible vs the 1.51 ms backbone decode. The observed +0.16 ms overhead is dominated by host-device synchronization and kernel-launch costs, not transfer bandwidth.
> - **Prefill:** ~400 unique tokens (after dedup from 512) × $d$ × 2 ≈ 600 KB per layer, ~10.8 MB across 18 layers, taking ~0.17 ms at PCIe Gen5, resulting in only +0.27 ms (+2.2%) end-to-end.
>
> > Key Q2. How do gains vary with token frequency and language?
>
> This question is closely related to W3; we expand here using the full 3.2B-A0.5B breakdown (Table 1, 14 benchmarks, overall avg +5.59).
>
> **Language.** C3 (Chinese reading comprehension) yields the single largest gain among all 14 benchmarks at **+12.17**. Chinese tokens are underrepresented in the multilingual corpus — precisely the "tail" the reviewer has in mind — yet this benchmark benefits the most.
>
> **Token frequency.** Tasks depending on rare vocabulary show substantial gains: TriviaQA **+6.72** (entity names), DROP **+6.58** (rare spans), GSM8K **+6.31** (math notation), MBPP **+5.31** (code tokens) — all comparable to or above the +5.59 overall average. Meanwhile, common-English benchmarks like Hellaswag (+3.32) and SocialIQA (+4.52) show the *smallest* gains. The pattern, if any, favors tail tokens rather than disadvantaging them.
>
> As noted in W3, $\mathrm{Norm}_\varepsilon$ ensures this by projecting modulation vectors to unit norm. The normalization ablation (Appendix B.2) confirms: removing it disproportionately hurts CMMLU and CEval (Chinese), validating that normalization is essential for uniform benefit across the frequency spectrum.
>
> [1]Huang, Hongzhi, et al. "Over-tokenized transformer: Vocabulary is generally worth scaling." arXiv preprint arXiv:2501.16975 (2025).

---

> > ### Author Rebuttal · Reviewer_T41c · 2026-04-01
> >
> > I appreciate the authors' effort and find their response satisfactory. I thank you for your engagement in the rebuttal.
> >
> > I have decided to raise my score.

---

> > > ### Author Response · Authors · 2026-04-03
> > >
> > > We sincerely thank the reviewer for the acknowledgment of our work and for raising the score. We appreciate the time and effort devoted to reviewing our paper.

---

### Official Review · Reviewer_oTQB · 2026-02-26

**Soundness:** 3
**Presentation:** 2
**Significance:** 3
**Originality:** 3
**Overall Recommendation:** 4
**Confidence:** 3

**Summary:**

This work proposes a new method for scaling the capacity of large language models. The approach is based on a learnable embedding table that can be optimized for system efficiency. ReToken and MoRT demonstrate strong effectiveness on top of diverse backbone architectures across a wide range of benchmarks.

**Compliance With Llm Reviewing Policy:**

Affirmed.

**Final Justification:**

My concerns have been addressed and I vote for acceptance.

**Key Questions For Authors:**

Can ReToken or MoRT achieve greater efficiency than MoE in model scaling?

**Limitations:**

The method is evaluated primarily on relatively small-scale experimental settings. (Note that I do not consider this a decisive factor in my evaluation.)

**Strengths And Weaknesses:**

(+) The method is clearly explained with figures and equations

(+) A lot of experiments are conducted to support the effectivess of the method, both pretraining/downstream and a lot of model scales.

(+) Using embedding tables to scaling the model capacity if genearl interesting and the effectivness is consistent.

(-) The claim is unclear to me, in the abstract, seems like the ReToken and MoRT is a better variant of scaling models than MoE and dense,
while the experiments in Figure 2 shows ReToken can further improve MoE and Dense as an orthogonal method. It would be clear to state the method is an additional direction to scaling models instead of better one than MoE.

(-) What is the relationship between the proposed method and other embedding table based method, like STEM (https://arxiv.org/pdf/2601.10639)

---

> ### Author Rebuttal · Authors · 2026-03-31
>
> > W1. Claim unclear: abstract implies better than MoE, but experiments show orthogonal/complementary.
>
> We agree that “orthogonal and complementary” is the right framing: our method augments both dense and MoE backbones rather than replacing them. We will revise the abstract to make this positioning clearer.
>
> > W2. Relationship with STEM (arXiv:2601.10639)?
>
> We thank the reviewer for pointing out STEM, which is a highly relevant concurrent work.
>
> The key difference is how the retrieved parameters interact with the backbone. STEM replaces the FFN up-projection with a layer-local token embedding while keeping the gate and down-projection unchanged. In contrast, ReToken/MoRT do not replace backbone projections: ReToken modulates the existing MLP residual with a token-indexed vector, and MoRT further extends this with hidden-state-conditioned routing over multiple token modulators.
>
> A second difference is context adaptivity. In STEM, the retrieved row is fixed by token identity, while contextual adaptation mainly comes from the preserved dense gate path. In MoRT, the selected modulator itself changes with the hidden state, so the same token can receive different modulation in different contexts.
>
> Finally, the emphasis is also different. STEM mainly highlights stability and interpretability of static token-indexed FFN replacement, whereas our work focuses on establishing token-indexed parameters as a principled scaling axis — demonstrating a 35% compute saving under isoFLOPs analysis and a predictable power-law scaling behavior.
>
> > Key Questions. Can ReToken/MoRT achieve greater efficiency than MoE in scaling?
>
> When the MoE backbone already operates at high sparsity, yes. Our 17B-A1.5B MoE backbone activates top-8 out of 144 experts, a regime where further expert scaling yields diminishing returns. We compare two ways of adding an identical 36B parameter budget: (a) expanding routed experts from 144 to 448, and (b) attaching 36B MoRT parameters ($n_e$=4, top-$K$=2). Backbone architecture, activated parameters, training data (87.5B tokens), and recipe are all identical.
>
> **Table R1: Downstream comparison (17B-A1.5B, 87.5B tokens)**
>
> | | Hellaswag | Xiezhi | CMMLU | MMLU | ARC-C | CEval | C3 | **Avg** |
> |---|---|---|---|---|---|---|---|---|
> | 17B backbone-only | 35.96 | 57.13 | 48.81 | 46.94 | 45.70 | 50.68 | 62.69 | 49.70 |
> | 17B + 36B MoE experts | 36.02 | 59.32 | 51.78 | 47.76 | 49.49 | 53.20 | 67.73 | 52.19 |
> | 17B + 36B MoRT | **39.93** | **61.98** | **56.00** | **50.49** | **54.30** | **55.15** | **71.51** | **55.62** |
>
> MoRT outperforms MoE expert scaling by **+3.43** avg across all seven benchmarks. Inference efficiency also favors MoRT (2×H800, SGLang, batch_size=16, seq_len=4096):
>
> **Table R2: Inference throughput (17B-A1.5B)**
>
> | | Prefill (tok/s) | Decode (tok/s) |
> |---|---|---|
> | 17B backbone-only | 113.1K | 1,812 |
> | 17B + MoE experts | 94.3K (-16.6%) | 1,589 (-12.3%) |
> | 17B + MoRT | 112.4K (-0.6%) | 1,693 (-6.6%) |
>
> MoRT delivers higher accuracy at better inference efficiency, as its token-indexed tables reside on CPU and avoid GPU memory pressure. This suggests that in the high-sparsity regime where MoE scaling saturates, token-indexed parameters are a more parameter-efficient scaling axis — while remaining complementary to MoE (i.e., they can be stacked on top of existing MoE backbones). We have not compared the two axes under low-sparsity regimes, and their relative efficiency in that setting remains an open question.
>
> > Limitations. Evaluated only on small-scale settings.
>
> We have extended our evaluation to a **17B-A1.5B MoE** backbone (87.5B tokens, $n_e$=4, top-$K$=2, 36B token-indexed parameters). As shown in Table R1 above, MoRT yields **+5.92** avg gain with consistent improvements across all seven benchmarks. We plan to extend validation to frontier-scale models in future work.

---

> > ### Author Rebuttal · Reviewer_oTQB · 2026-04-03
> >
> > Thank you for the thorough responses. I am satisfied with the clarifications provided and am happy to vote for acceptance.

---

> > > ### Author Response · Authors · 2026-04-03
> > >
> > > We sincerely thank the reviewer for the positive evaluation and for acknowledging our responses.  Given that all concerns have been fully resolved, would you consider raising your overall score accordingly?
> > > Thank you again for your time and constructive feedback.

---

### Official Review · Reviewer_mRMa · 2026-03-08

**Soundness:** 4
**Presentation:** 3
**Significance:** 3
**Originality:** 3
**Overall Recommendation:** 4
**Confidence:** 3

**Summary:**

The authors explored a new dimension of LLM scaling: Token-Index Parameters.
They demonstrate that model performance and FLOPs can potentially be decoupled.
By adding ReToken and MoRT to the Transformer backbone, they achieved significant performance improvements with a minimal increase in FLOPs.
Furthermore, they trained a wide range of models on both dense and MoE using their method and researched the scaling laws for them, discovering that they obtain a better Pareto frontier under the same FLOPs budget.

**Compliance With Llm Reviewing Policy:**

Affirmed.

**Final Justification:**

The authors conducted very detailed experiments to address the issues I raised, so I believe the soundness of the paper has improved.

**Key Questions For Authors:**

See weaknesses.

**Limitations:**

The authors did not explicitly discuss their limitations in the article. It would be better if it is discussed.

**Strengths And Weaknesses:**

Strengths:
1. The paper is overall good-written, particularly the related works section is detailed and comprehensive.
2. The scaling angle (token-indexed parameters) in the article explores an area that is under-explored, making a new contribution to the existing scaling laws.
3. The authors conducted experiments across a wide range of models, including dense and MoE models. They found that their method significantly improves performance across various model sizes.

Weaknesses:
1. The paper does not explicitly state the additional parameter overhead for each trained model. Based on my rough estimation, the ReToken parameters for the small model (190M parameters) is about 500M. Please provide a detailed breakdown of the extra parameter count for each model.
2. All performance comparisons are under the condition of same backbone (but with different number of parameters). In order to ensure a fair comparison and to gain a more detailed understanding of the proposed method, I would like to see performance comparison under the same number of parameters or the same (or approximately the same) number of activated parameters.
3. The model architecture lacks ablation studies. For instance, why is the ReToken placed after the MLP rather than after the attention layer? Why does the MoRT router receive activations from before the attention layer?
Due to the limited time and resources during rebuttal, if it's hard to conduct comprehensive experiments, please provide some empirical explanations instead.
4. Although the article mentions the load balancing issue in MoRT and introduces a balancing loss, the authors did not perform an ablation study on this loss. It will be better to show the actual load balancing status with / without balancing loss.
5. Although the authors used various methods to improve the efficiency of both training and inference, I still have some concerns. The paper provides throughput for training and inference, but these metrics are based on a fixed batch size (correct me if I'm wrong). In real-world production environment, inference is typically streaming; requests should saturate the GPU memory rather than being confined to a fixed batch size. So I would like to see a comparison of inference throughput when the batch size is maximized with the available GPU memory and / or a comparison of GPU (and CPU) memory usage with / without ReToken when using the same batch size.
6. Could authors discuss the relationship between this work and recent N-gram-based lookup tables? What are the advantages of your method?
    - Cheng X, Zeng W, Dai D, et al. Conditional memory via scalable lookup: A new axis of sparsity for large language models[J]. arXiv preprint arXiv:2601.07372, 2026.
    - Liu H, Zhang J, Wang C, et al. Scaling Embeddings Outperforms Scaling Experts in Language Models[J]. arXiv preprint arXiv:2601.21204, 2026.

Minor issues / typos:
1. If the cited article serves as the subject of the sentence, I recommend using the \citet command, which is the version without parentheses.
2. The formula in the left column of line 155 misses a "max": $\mathcal{L}^*(C) = \max \mathcal{L}(N_c, D)$
3. In the right column from lines 149 to 152, some symbols change between bold and regular.
4. On the right side of line 160, the subtitle "Token-modulated Mixture of MoE" should be "Token-Modulated Mixture of ReToken"
5. The two paragraphs starting in the left column on line 215 should be capitalized at the beginning.

---

> ### Author Rebuttal · Authors · 2026-03-31
>
> > W1. Parameter overhead not stated; provide breakdown per model.
>
> Per layer: $V \times d$ for ReToken, $n_e \times V \times d$ for MoRT. Full breakdown:
>
> | Model | Backbone | $V$ | +ReToken | +MoRT ($n_e$=5) |
> |---|---|---|---|---|
> | Dense-S | 190M | 50,304 | 0.46B | — |
> | Dense-M | 505M | 50,304 | 1.24B | — |
> | Dense-L | 1.0B | 50,304 | 2.32B | — |
> | Dense-XL | 1.5B | 152,064 | 6.54B | — |
> | 1.5B-A250M MoE | 1.5B | 152,064 | 0.93B | 4.67B |
> | 3.2B-A0.5B MoE | 3.2B | 152,064 | 2.10B | 10.51B |
>
> > W2. Need comparison under matched total or activated parameters.
>
> We add identical 36B params to 17B-A1.5B two ways — MoE expert scaling (144 to 448) vs. MoRT — with backbone, activated params, and training all fixed. MoRT outperforms by +3.43 avg (55.62 vs. 52.19), confirming token-indexed parameters are more efficient under matched total and activated parameters. Please see the rebuttal to Reviewer u4kb-W4 for details.
>
> > W3. Missing ablations: why ReToken after MLP? Why router uses pre-attn input?
>
> Ablations on 17B-A1.5B (87.5B tokens, identical settings):
>
> **ReToken placement: after MLP vs. after attention.**
>
> | | Hellaswag | Xiezhi | CMMLU | MMLU | ARC-C | CEval | C3 | **Avg** |
> |---|---|---|---|---|---|---|---|---|
> | Backbone-only | 35.96 | 57.13 | 48.81 | 46.94 | 45.70 | 50.68 | 62.69 | 49.70 |
> | After attn | 34.41 | 58.98 | 49.40 | 46.95 | 48.45 | 51.20 | 64.11 | 50.50 |
> | After MLP (default) | 36.71 | 59.02 | 51.88 | 48.56 | 49.14 | 50.96 | 64.77 | **51.58** |
>
> After-MLP (+1.88 avg) outperforms after-attn (+0.80, even degrades Hellaswag). FFN outputs encode stable feature patterns amenable to token-type calibration; attention outputs are dynamically composed contextual messages where static rescaling may distort context integration.
>
> **MoRT router input: pre-attention vs. pre-MLP.**
>
> | | Hellaswag | Xiezhi | CMMLU | MMLU | ARC-C | CEval | C3 | **Avg** |
> |---|---|---|---|---|---|---|---|---|
> | Router at MLP input | 41.04 | 61.43 | 53.71 | 49.44 | 50.62 | 56.13 | 68.77 | 54.45 |
> | Router at attn input (default) | 39.93 | 61.98 | 56.00 | 50.49 | 54.30 | 55.15 | 71.51 | **55.62** |
>
> Pre-attention outperforms pre-MLP by +1.17 avg. Both inputs carry context from preceding layers; they differ only in whether the current layer's attention has been applied. Pre-attention also provides a system benefit: larger overlap window between MoRT and backbone (attention + MLP).
>
> > W4. No ablation on load-balancing loss; show actual load status.
>
> Ablation on 17B-A1.5B (36B MoRT, $n_e$=4, top-$K$=2, 87.5B tokens, LBL $\lambda$=1e-4):
>
> | | Hellaswag | Xiezhi | CMMLU | MMLU | ARC-C | CEval | C3 | **Avg** |
> |---|---|---|---|---|---|---|---|---|
> | With LBL | 39.93 | 61.98 | 56.00 | 50.49 | 54.30 | 55.15 | 71.51 | **55.62** |
> | No LBL | 38.68 | 57.34 | 51.23 | 48.60 | 47.77 | 52.47 | 65.70 | 51.68 |
>
> Removing LBL causes **−3.94** avg drop. The metric $n_e \sum_i p_i f_i$ (Eq. 27) converges to ~1.98 without LBL vs. ~1.22 with LBL (near-optimal 1.0), confirming imbalanced routing starves experts of training signal.
>
> > W5. Inference throughput at max batch; GPU/CPU memory comparison.
>
> Memory comparison on 3.2B-A0.5B (1×H800, bs=16, seq=4096, bf16):
>
> | | peak GPU HBM | CPU Host Memory |
> |---|---|---|
> | Backbone-only | 7.8 GB | — |
> | +ReToken (CPU offload) | 7.9 GB | +4.2 GB |
>
> With CPU offloading, token-indexed tables reside in host memory. The only GPU overhead is a receive buffer (~0.1GB for prefill, negligible for decode). CPU memory cost ≈ ReToken size in bf16 (2.10B × 2 bytes = 4.2 GB).
>
> > W6. Relationship with N-gram methods (Engram, LongCat)?
>
> Engram and LongCat are concurrent works with the same high-level motivation. Three key differences:
>
> (1) **Lookup.** They use hash-based N-gram lookup keyed by local contexts; we use exact token-ID lookup, avoiding hash collisions and extra design choices (N-gram order, hash-table sizing).
>
> (2) **Context adaptivity.** LongCat retrieves by local token history; Engram adds a scalar gate from hidden states to reweight the retrieved embedding. MoRT is more adaptive: its router dynamically selects and mixes from multiple modulator pools, producing different modulation for the same token in different contexts.
>
> (3) **Integration.** Our method is natively per-layer and modulates the Transformer block directly. LongCat's per-layer extension (PLNE), the closest to ours, was not adopted in their main model as it increases activated params without consistent advantage.
>
> We view these as complementary: Engram/LongCat emphasize local-pattern memory; ReToken/MoRT provide per-layer modulation.
>
> > Limitations: No limitations section.
>
> Main limitations: (1) Backbone scale is relatively small and frontier-scale validation remains future work; (2) evaluation focuses on pretraining; interaction with post-training is unexplored; (3) interaction of token-indexed parameters with inference compression (quantization, distillation) is not studied. Will add a limitations section.

---

> > ### Author Rebuttal · Reviewer_mRMa · 2026-04-02
> >
> > I would like to thank the authors for their comprehensive and detailed rebuttal. The responses have effectively addressed the majority of my initial concerns. Given the thoroughness of the clarification provided, I am pleased to raise my score for Soundness to 4.

---

> > > ### Author Response · Authors · 2026-04-03
> > >
> > > We sincerely thank the reviewer for the thorough and constructive evaluation. We are grateful that our responses have resolved your concerns, and we appreciate your recognition of our contributions.

---

### Official Review · Reviewer_u4kb · 2026-03-12

**Soundness:** 4
**Presentation:** 3
**Significance:** 3
**Originality:** 3
**Overall Recommendation:** 4
**Confidence:** 2

**Summary:**

This paper proposes token-indexed parameters as a new scaling axis for LLMs. By introducing ReToken and MoRT, the authors use lightweight element-wise modulation to increase model capacity with negligible FLOPs overhead. Experiments show a 35% compute saving and significant gains across dense and MoE models up to 9.8B parameters.

**Compliance With Llm Reviewing Policy:**

Affirmed.

**Final Justification:**

It has addressed my main concerns.

**Key Questions For Authors:**

Please refer to the weaknesses listed above.

**Limitations:**

yes

**Strengths And Weaknesses:**

Strengths：

1.The paper addresses a highly topical and critical challenge in the field—decoupling model capacity from computational cost to overcome the limitations of traditional dense scaling.

2.The methodology is exceptionally efficient. By utilizing element-wise modulation instead of dense matrix multiplications, the added computational cost is only $\mathcal{O}(d)$ per layer, which is significantly lower than the $\Theta(d^{2})$ complexity of the backbone’s attention and FFN components.

3.The authors provide a robust empirical evaluation across various architectures, including both dense and Mixture-of-Experts backbones. The proposed methods demonstrate significant gains across multiple downstream domains such as knowledge, coding, mathematics, and reasoning.

Weaknesses：

1.The token-indexed tables may significantly increase the memory overhead. Although the authors mention CPU offloading as a mitigation strategy, the discussion regarding memory bottlenecks should be deepened for scenarios involving ultra-large vocabulary sizes ($V$) or an increased number of experts.

2.The current experiments primarily focus on standard sequence lengths. Given that ReToken relies on token-specific retrieval, its impact on attention consistency and long-range dependency modeling within extremely long context windows remains a critical area that warrants further investigation.

3.The core results presented in Table 1 are limited to models $\le$ 3.2B parameters. It remains unclear how these performance gains，specifically regarding accuracy, computational efficiency, and memory overhead，will translate to significantly larger models, such as 7B, 70B, or beyond.

4.The paper positions token-indexed parameters as a "novel scaling axis," but it lacks a rigorous direct comparison (accuracy, computational efficiency, and memory overhead) with traditional depth scaling within the same architecture.
Quantify the parameter and compute savings of ReToken relative to standard baselines at iso-accuracy.

---

> ### Author Rebuttal · Authors · 2026-03-31
>
> > W1. Memory overhead for ultra-large V or n_e.
>
> Token-indexed parameters per layer total $n_e \times V \times d$, scaling linearly with $V$ and $n_e$. We discuss training and inference with new 17B-A1.5B experiments.
>
> **Training.** Per-GPU memory stays constant via emb parallelel: tables are sharded row-wise, scaling embP group size to keep per-GPU params fixed. Owner-rank premixing bounds communication at $O(d)$ regardless of table size. Larger embP groups add latency-bound collectives, slightly decreasing overlap efficiency.
>
> **Table R1** (17B-A1.5B, 128×H800, top-K=2)
>
> | MoRT Params | EmbP Size | Per-GPU Emb Params | Throughput | vs. Backbone |
> |:---:|:---:|:---:|:---:|:---:|
> | (backbone only) | — | 0 | 1,748K tok/s | 100% |
> | 18B | 4 | 4.5B | 1,624K | 92.9% |
> | 36B | 8 | 4.5B | 1,601K | 91.6% |
> | 72B | 16 | 4.5B | 1,574K | 90.0% |
>
> Per-GPU params stay at 4.5B. Quadrupling capacity (18B→72B) costs only ~3% more throughput, less than half MoRT's 7.1% base cost.
>
> **Inference.** CPU offloading (Sec 3.4.5) places tables in host memory, adding zero GPU HBM overhead regardless of $V$ or $n_e$. Transfer volume depends only on batch tokens (top-K * $d$), not table size.
>
> **Table R2** (17B-A1.5B, 2×H800, bs=16, seq=4096)
>
> |  | Backbone-only | Emb 18B | Emb 36B | Emb 72B |
> |:---|:---:|:---:|:---:|:---:|
> | Prefill (tok/s) | 113.1K | 112.5K | 112.4K | 112.3K |
> | Decode (tok/s) | 1,812 | 1,697 | 1,693 | 1,693 |
>
> 18B→72B shows no measurable throughput change, confirming independence from table size.
>
> > W2. Impact on attention consistency and long-range modeling.
>
> ReToken/MoRT do not interact with attention: ReToken modulates only the MLP residual; MoRT injects an additive residual alongside MLP output. Long-range dependencies are handled by unchanged attention layers; consistency is preserved by construction.
>
> Second, the modulation is length-agnostic: ReToken depends only on token id, not position or length. MoRT's router operates on hidden states carrying long-range context from preceding layers, making selection context-aware without position-dependent mechanisms.
>
> Besides, our experiments use 8192 seq_len for MoE and Dense-XL (Appendix Tables 4–5), the standard length of recent LLMs, with consistent gains confirming the method works beyond short sequences.
>
> > W3. Results limited to ≤3.2B; gains at 7B/70B+ unclear.
>
>  Our scaling experiments already extend well beyond 3.2B: Figure 3 covers up to 9.8B-A1.4B, showing stable improvements across the range of 2.3B to 9.8B.
>
> We further trained a 17B-A1.5B MoE with 36B MoRT params ($n_e$=4, top-K=2) with 87.5B tokens (in compute optimal setting). MoRT achieves **+5.92** avg gain across seven benchmarks (Table R3), confirming substantial gains at larger scale.
>
> While experiments at the 70B+ scale are beyond our current compute budget, three converging lines of evidence support that gains will carry over: (a) Fig. 3 shows stable relative loss improvements across a range of 2.3B to 9.8B; (b) 17B results show significant downstream gains persist at larger scale; (c) Eq. 17 showing compute saving is scale-independent.
>
> Efficiency at 17B-MoRT: see Tables R1–R2.
>
> > W4. Lacks direct comparison with traditional scaling axes at iso-accuracy.
>
> We conduct a controlled head-to-head comparison between scaling axes. In the MoE setting, adding layers would increase activated FLOPs, breaking the iso-FLOPs condition. The standard approach to scale MoE capacity at fixed activated FLOPs is to increase routed experts while keeping top-K fixed [1]. Thus we add identical 36B params to 17B-A1.5B two ways: (a) expand routed experts 144 to 448, (b) attach MoRT ($n_e$=4, top-K=2).  Activated params, training data and recipe are identical.
>
> **Table R3** (17B-A1.5B, 87.5B tokens)
>
> | | Hellaswag | Xiezhi | CMMLU | MMLU | ARC-C | CEval | C3 | **Avg** |
> |---|---|---|---|---|---|---|---|---|
> | 17B backbone | 35.96 | 57.13 | 48.81 | 46.94 | 45.70 | 50.68 | 62.69 | 49.70 |
> | +36B MoE experts | 36.02 | 59.32 | 51.78 | 47.76 | 49.49 | 53.20 | 67.73 | 52.19 |
> | +36B MoRT | 39.93 | 61.98 | 56.00 | 50.49 | 54.30 | 55.15 | 71.51 | 55.62 |
>
> MoRT outperforms MoE expert scaling by **+3.43** avg.
>
> Efficiency comparison (2×H800, SGLang, bs=16, seq=4096):
>
> **Table R4** (17B-A1.5B)
>
> | | Prefill (tok/s) | Decode (tok/s) |
> |---|---|---|
> | 17B backbone | 113.1K | 1,812 |
> | +MoE experts | 94.3K (−16.6%) | 1,589 (−12.3%) |
> | +MoRT | 112.4K (−0.6%) | 1,693 (−6.6%) |
>
> MoE expert scaling reduces prefill throughput by 16.6% and decode by 12.3%, as experts must reside in GPU. MoRT, by contrast, incurs only 0.6% prefill and 6.6% decode overhead via CPU offloading (see W1, Table R2). This means MoRT delivers higher accuracy at better inference efficiency. This aligns with Sec. 3.5: at high sparsity (top-8/144), expert scaling has limited headroom, while token-indexed modulation uses a complementary channel.
>
> [1] Krajewski et al., arXiv:2402.07871.
> [2] Tian et al., arXiv:2507.17702.

---

> > ### Author Rebuttal · Reviewer_u4kb · 2026-04-03
> >
> > I would like to thank the authors for their detailed response. I have decided to increase my Soundness score to 4.

---

> > > ### Author Response · Authors · 2026-04-03
> > >
> > > We sincerely thank the reviewer for the time and effort in evaluating our work. We are glad that our responses have addressed your concerns, and we appreciate your recognition of our contributions.

---

### Decision · Program_Chairs · 2026-04-30

**Decision:**

Accept (regular)

**Comment:**

This paper proposes a simple approach for scaling Transformers based on embedding tables that modulate intermediate computations within Transformers.  This increases the number of parameters but keeps the FLOPs and memory movement (and hence training/inference latency) roughly the same, resulting in little-to-no latency overhead. Across both dense and MoE architectures, as well as model scales (up to 1B dense and 3.2B MoE trained on 100B tokens), the approach soundly outperforms baseline methods.

While the novelty of the layer itself is not super high (given works such as Engram, LongCat, STEM, some of which are concurrent), the careful empirical study, combined with scaling laws, makes this work a solid contribution to ICML.